# Enhanced and unified anatomical labeling for a common mouse brain atlas

Uree Chon [1], Daniel J. Vanselow [2], Keith C. Cheng [2] & Yongsoo Kim [1]*

Anatomical atlases in standard coordinates are necessary for the interpretation and integration of research findings in a common spatial context. However, the two most-used mouse brain atlases, the Franklin-Paxinos (FP) and the common coordinate framework (CCF) from the Allen Institute for Brain Science, have accumulated inconsistencies in anatomical delineations and nomenclature, creating confusion among neuroscientists. To overcome these issues, we adopt here the FP labels into the CCF to merge the labels in the single atlas framework. We use cell type-specific transgenic mice and an MRI atlas to adjust and further segment our labels. Moreover, detailed segmentations are added to the dorsal striatum using cortico-striatal connectivity data. Lastly, we digitize our anatomical labels based on the Allen ontology, create a web-interface for visualization, and provide tools for comprehensive comparisons between the CCF and FP labels. Our open-source labels signify a key step towards a unified mouse brain atlas.

[1] Department of Neural and Behavioral Sciences, College of Medicine, Penn State University, Hershey, PA, USA. [2] Department of Pathology, College of Medicine, Penn State University, Hershey, PA, USA. *email: yuk17@psu.edu

Anatomical delineation of the brain is critical for elucidation of the anatomical and functional organization of the brain across species[1–5]. Whole brain anatomical atlases provide a spatial framework for examining, interpreting, and comparing experimental data from different studies. For the mouse, the most widely used animal model to understand the mammalian brain, a variety of printed and/or digital atlases exist with varying levels of segmentations in 2D or 3D images acquired from different imaging modalities (e.g., Nissl staining or MRI)[6–11]. Among many, the Franklin-Paxinos atlas (FP)[6] and the Allen Reference Atlas (ARA)[7,8] are the two most commonly used brain atlases[12,13]. Both atlases are largely based on manual delineation by expert neuroanatomists using cytoarchitectonic features based on a variety of staining including Nissl and acetylcholine esterase antibody staining in 2D histological sections.

In 2015, the Allen Institute for Brain Sciences released a 3D reference brain with 10 μm isotropic voxel resolution, called the Allen Common Coordinate Framework (Allen CCF)[14]. This reference brain marked a significant departure from classical neuroanatomy based on 2D sections and provides an excellent platform for the registration of 3D mouse brain imaging datasets collected from in vivo imaging (e.g., PET, MRI) and emerging high-resolution whole-brain imaging modalities such as serial two-photon tomography and light sheet fluorescent microscopy[5,15–18]. More importantly, the Allen CCF facilitates the integration and sharing of scientific data from different studies in a common spatial context[19]. The accompanying anatomical labels have smooth delineation across all 3D planes, which enable easy views of 3D perspective of brain regions.

Unfortunately, significant discrepancies exist between the anatomical labels on the ARA and the FP labels. For example, these two atlases often have discordant anatomical borders and 3D coordinates as well as different names for the same structures[12,20]. To make it worse, the labels in the Allen CCF released in 2017 (CCFv3) also introduced significant changes from its original ARA labels that were based on 2D Nissl stained sections. This has created confusion and misinterpretation of experimental results[21]. These issues motivated us to create a unified and highly segmented anatomical labeling system in the adult mouse brain based on the Allen CCF. We decided to use the FP labels for our initial anatomical labeling because it represents one of the most popular adult mouse brain atlases with detailed segmentations, and because a huge body of prior research is based on the FP labels[12,18]. Here, we adopt the FP labels into the Allen CCF by rigorous alignment using an MRI based atlas and cell type specific transgenic mice marking for distinct anatomical areas[18,22]. We also further segment labels where cell types could be distinguished within single anatomically defined regions. The resulting labels create a unique opportunity for comprehensive comparisons between the two most frequently used anatomical labels in a common space. Furthermore, we use topographically distinct cortico-striatal projection patterns to add segmentations to the dorsal striatum, which is unsegmented in the existing atlases.

Lastly, we digitize the anatomical labels based on the Allen ontology to facilitate integration of labels as a neuroinformatics tool[14]. Digitized labels combined with image registration can serve as a powerful tool to automatically quantify signal of interest across whole brain regions in a reference brain[14,15,23,24]. To facilitate its usage, our digital map data is freely available for viewing and downloading from our web-based atlas implementation at http://kimlab.io/brain-map/atlas/.

## Results

**Importing FP anatomical labels into the Allen CCF.** We used the FP labels drawn in 2D histological sections for our initial template segmentation. We first imported vector drawings of FP labels into the Allen CCF (Fig. 1a, b). Automated image registration of 2D Nissl sections from the FP atlas to the Allen CCF has been challenging due to differences in background content between the two atlases and non-uniform tissue distortion between histological sections in the FP labels. Thus, we used manual adjustment to initially align the FP labels on the Allen CCF coronal sections with 100-μm z spacing based on the autofluorescence signals of distinct anatomical features (Fig. 1b, c, yellow arrows as examples). Autofluorescent background in the Allen CCF provides rich anatomical information in both cortical and subcortical regions. For example, distinct contrast in the barrel field enables the delineation of layer 4 of the somatosensory barrel cortex (Fig. 1b, c, red arrows).

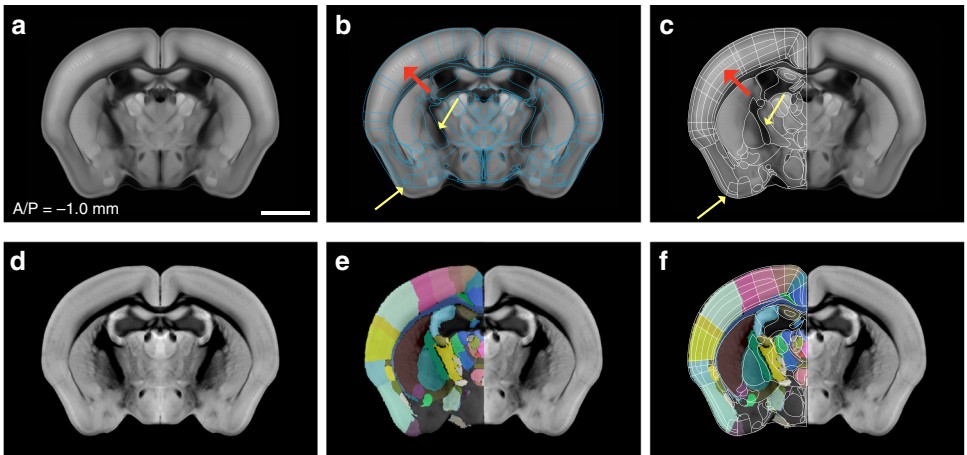

**Fig. 1** Import and alignment of the FP labels onto the Allen CCF. **a** The Allen Common Coordinate Framework (CCF) that serves as base anatomical platform. A/P represent Bregma anterior/posterior coordinates. **b** Initial import of the Franklin-Paxinos (FP) vector labels into the Allen CCF. **c** Manual alignment based on anatomical features in the Allen CCF. Yellow arrows highlight distinct anatomical boundaries based on edges and white matter tracks. Red arrows indicate layer 4 in the somatosensory barrel cortex. **d** MRI images registered to the same CCF plane in (**a**). **e** Original FP based labels drawn in the MRI atlas registered to the Allen CCF. The lack of labels in the hypothalamic and amygdala regions are due to missing labels in the original MRI annotation. **f** Further adjustment of FP based anatomical delineation (white lines) based on the MRI labels. Scale bar = 2 mm

To further assist 2D label alignment in the context of contiguous 3D planes, we used a high-resolution magnetic resonance imaging (MRI) atlas with the FP labels in most brain regions[18,25,26]. We first registered the MRI reference brain to the Allen CCF and transformed the MRI labels to fit in the CCF (Fig. 1d, e). Although the MRI labels are not as detailed as Nissl based FP labels, they provide an independent way to align and to further adjust our initial alignment in 3D space (Fig. 1f). The MRI labels were particularly useful for aligning segmentations in the isocortex (also called "neocortex") (Fig. 1f).

**Refining labels using cell type-specific transgene expression.** Previously, histological staining with specific markers (e.g., acetylcholine esterase, or parvalbumin) on 2D sections has been used to guide detailed delineation in anatomical regions[6]. We utilized a similar approach using 14 different transgenic mouse lines that mark specific neuronal subtypes[15,22,27] (called marker brains). We chose marker brains from different neuropeptides, neurotransmitters, transcription factors, G protein coupled receptors, calcium binding proteins, and a growth factor that highlight anatomical boundaries otherwise often not visible in the Allen CCF tissue autofluorescent background (Supplementary Data 1). Marker brains imaged by STPT were registered to the Allen CCF, and their signals were overlaid in the Allen CCF to highlight cell type based anatomical features (Fig. 2, Supplementary Figs. 1 and 2, Supplementary Data 1). For example, Choline acetyltransferase (Chat)-Cre mice crossed with Cre-dependent reporter mice expressing nuclear tdTomato (Ai75) were used to delineate brain regions enriched with cholinergic neurons such as the basal forebrain and the hindbrain areas (Fig. 2a–c)[28]. Parvalbumin (PV)-Cre crossed with Cre dependent reporter mice expressing nuclear GFP (H2B-GFP) were useful for delineating structures in the thalamus, midbrain, and hindbrain (Fig. 2d–f)[6,29]. Somatostatin (SST)-Cre crossed with H2B-GFP reporter mice have been useful for amygdala, hypothalamus, olfactory regions, and subcortical regions, such as the bed nucleus of the stria terminalis (BST) (Fig. 2g–i, Supplementary Fig. 2)[30]. Oxytocin receptor (OTR)-Cre crossed with Cre dependent reporter mice expressing tdTomato (Ai14) highlighted selected brain regions including dorsal endopiriform nucleus (DEn), CA2 in the hippocampus, amygdala, and entorhinal regions (Fig. 2j–l)[31]. Lastly, we used cortical layer specific Cre mice crossed with Ai75 to validate our cortical layer (L) delineation. We used Ctgf-Cre for L6b, Ntsr1-Cre for L6, Rbp4-Cre for L5, and Cux2-Cre for L2/3 (Fig. 2m–o, Supplementary Fig. 1)[32]. Additional marker brains were utilized to delineate several more brain regions. For example, Ctgf-Cre was further used for delineations of DEn and structures of thalamus, amygdala, hypothalamus, and isocortical areas (Supplementary Figs. 1 and 2). The full list of marker brains and their expression in anatomical regions is summarized in Supplementary Data 1.

While utilizing marker brains, distinct cell populations were observed within specific substructures. Previous studies used gene expression patterns to delineate the different thalamic nuclei[33,34]. We have used this approach to further segment structures in the thalamus, hypothalamus, and hindbrain. Using PV-Cre and Cux2-Cre marker brains, the ventral posteromedial nucleus of the thalamus (VPM) was further segmented into dorsal and ventral parts (VPMd and VPMv, respectively) (Fig. 3a–d). We observed densely packed cell population in VPMd in both lines, contrasting the loosely scattered cells in VPMv (yellow arrows in Fig. 3a–d). We further examined whether VPM subdivisions created here are supported by differential neural connectivity, using the Allen Mouse connectivity database[24]. Indeed, VPMd and VPMv preferentially received inputs from anterior and posterior cortical

area, respectively (Supplementary Fig. 4). Similarly, utilizing the OTR-Cre and Ctgf-Cre marker brains, posterior hypothalamic nucleus (PH) was segmented into nuclear dorsal and ventral parts (PHnd and PHnv, respectively) with higher expression in PHnd (Fig. 3e–h). The laterodorsal tegmental nucleus, dorsal part (LDTg) was further segmented into lateral and medial divisions (LDTg-dl and LDTg-dm, respectively) using SST-Cre and PV-Cre marker brains (Fig. 3i–l), where higher expression is present in the LDTg-dl. The Barrington nucleus (Bar) was also further segmented into dorsal and ventral parts (Bard and Barv, respectively) using Chat-Cre and SST-Cre marker brains (Fig. 3m–p). Lastly, the medial vestibular nucleus, parvicellular part (MVp) was further divided into dorsal and ventral parts (MVpd and MVpv, respectively) based on density difference from SST-Cre and PV-Cre marker brains (Fig. 3q–t). Overall, transgene markers allowed us to add 10 subdivisions (Supplementary Data 2, Supplementary Fig. 3).

**Detailed anatomical segmentations in the caudate putamen.** Previously, anatomical segmentations were largely based on cytoarchitectonic features[6,7]. Although highly useful, this approach cannot be applied to the caudate putamen (CP, also called the dorsal striatum) without such features. Thus, the CP remains unsegmented in FP, ARA, and CCFv3 atlases despite its prominent size and heterogeneous functions in the brain. Recent studies have shown that different parts of the CP receive topographically distinct cortical inputs[35–37]. To confirm previous observations, we downloaded 129 datasets with anterograde tracing using C57bl/6 mice covering the entire isocortical area from the Allen connectivity dataset[24] and registered all of these brains to the Allen CCF (Fig. 4a–d). Then, we averaged the projection datasets from the 10 different cortical regions for each anatomically distinct CP projection pattern (Fig. 4e). We observed different striatal regions with either distinct input from one cortical group or convergent inputs from multiple regions (Fig. 4f), which is consistent with previous studies[35,36].

A recent study from the Mouse Connectome Project generated highly detailed CP segmentations based on discreet corticostriatal projections in the ARA[35]. We primarily used this dataset as well as our data from the Allen connectivity and another cortico-striatum projectome dataset from Hunnicutt et al.[36], to finely segment the CP. In the anterior-posterior axis, the CP was divided into the rostral extreme (re, Bregma A/P between +1.8 and +1.3), rostral (r, +1.2 and +0.7), intermediate (i, +0.6 and −0.4), caudal (c, −0.5 and −1.8), and caudal extreme (ce, −1.9 and −2.4). The CP at each level was further subdivided by community and domain as sub-segmentations as originally proposed by Hintiryan et al.[35]. For example, the CPi, dm, dl represents the dorsolateral (dl) domain within the dorsomedial (dm) community in an intermediate level CP (red arrow in Fig. 4j). We have added these delineations to the existing labels (Fig. 4g–n, Supplementary Data 2).

**Digitization of hierarchically organized anatomical labels.** Digital atlases with distinct label values for each anatomical region have been very useful neuroinformatics tools to automatically quantify target signals in different anatomical regions when combined with image registration[15,23]. To facilitate such efforts, we assigned a unique ID in each label (Fig. 5a–c). We adopted and arranged numerical IDs for each structure in a hierarchical manner based on the Allen ontology (Fig. 5f)[8,38]. In the digitization process, we first identified correspondences between the FP and the CCFv3 labels. To accommodate the higher degree of segmentation in our labels, 501 more structure IDs were created (Supplementary Data 2). For example, PAG

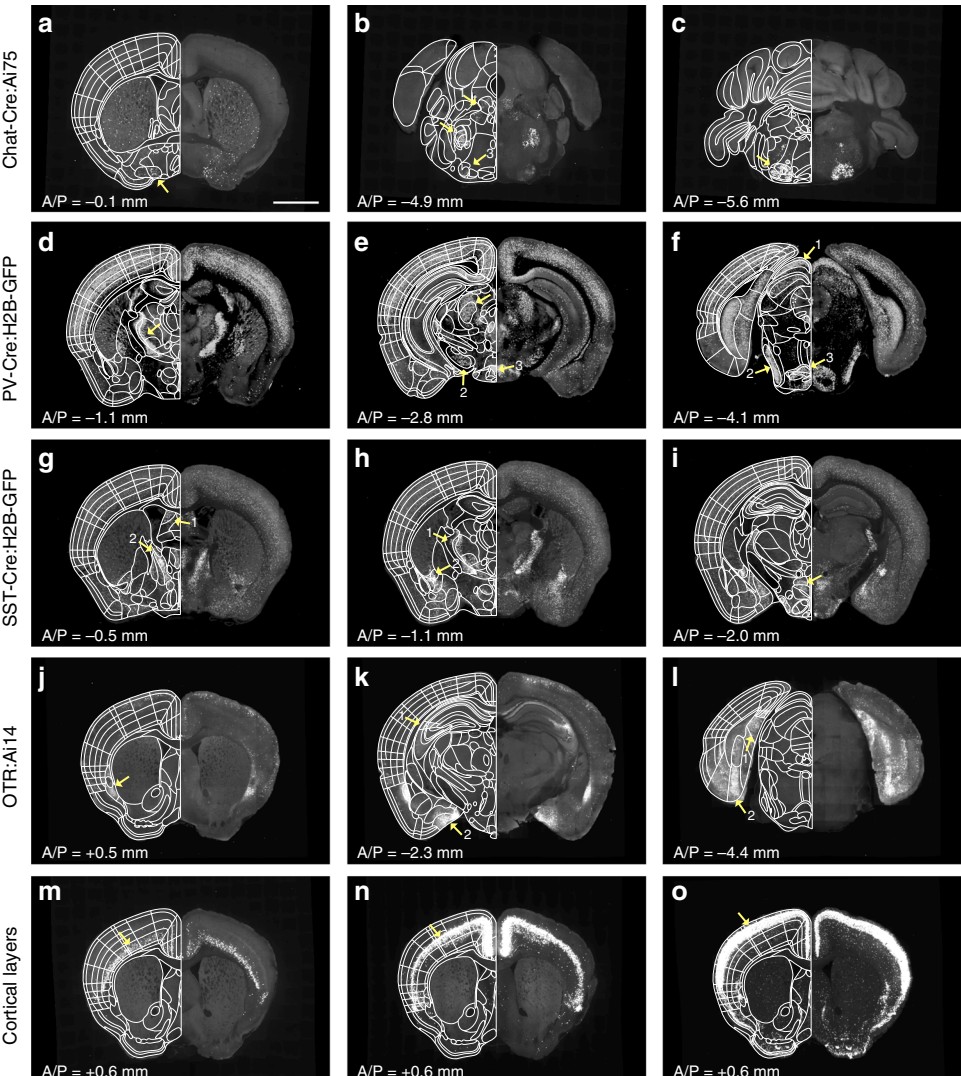

**Fig. 2** Marker brains for further alignment of anatomical labels. **a–o** Examples of different marker brains registered to the Allen CCF that helped to align FP based labels in subregions as highlighted with yellow arrows. A/P represents Bregma anterior/posterior coordinates. **a–c** Chat-Cre:Ai75 brain to delineate (**a**) the basal forebrain structures including the nucleus of the horizontal limb of the diagonal band (arrow). It was also used to delineate (**b**) midbrain areas, including the laterodorsal tegmental nucleus (arrow1), the motor trigeminal nuclei (arrow 2), the lateral superior olive (arrow 3), and **c** the facial nucleus (arrow). **d–f** PV-Cre:H2B-GFP brain to delineate (**d**) the reticular nucleus (arrow), **e** the anterior pretectal nucleus (arrow 1), the substantia nigra, reticular part (arrow 2), and the retromamillary nucleus (arrow 3) as well as **f** the superficial gray layer superior colliculus (arrow 1), the ventral nucleus of the lateral lemniscus (arrow 2), and the reticulotegmental nucleus of the pons, pericentral part (arrow 3). **g–i** SST-Cre:H2B-GFP brain to delineate (**g**) the cerebral nuclei, such as the lateral septal nucleus, dorsal part (arrow 1) and the bed nuclei of the stria terminalis medial division posteromedial part (arrow 2), **h** the reticular nucleus (arrow 1) and the central amygdaloid nuclei (arrow 2), and **i** hypothalamic structures, such as the dorsomedial hypothalamic nuclei dorsal and ventral parts (arrow). **j–l** OTR:Ai14 brain to delineate (**j**) the dorsal endopiriform nucleus (arrow), **k** CA2 (arrow 1), the posteromedial cortical amygdala (arrow 2), and **l** the caudomedial entorhinal cortex (arrow1) as well as the postsubiculum (arrow 2). **m–o** Cortical layers defined by **m** Ntsr-Cre:Ai75 for layer 6, **n** Rbp4-Cre:Ai75 for layer 5, and **o** Cux2-Cre:Ai75 for layer 2/3. Scale bar = 2 mm

consists of several subdivisions that play various functions including the expression of fear behavior[39]. PAG, which is considered a single structure in the CCFv3 labels, is further segmented into dorsomedial, lateral, dorsolateral, ventrolateral, pleoglial, and p1 divisions (DMPAG, LPAG, DLPAG, VLPAG, PlPAG, and p1PAG, respectively) in FP labels. Boundaries of the subdivisions were delineated by observing cell density differences between each division with SST-Cre expression (Fig. 5d). Anatomical connectivity data also showed that these subdivisions receive topographically distinct inputs from other brain regions (Supplementary Fig. 5). Each subdivided region was given a unique numerical ID and assigned within its parent structures (Fig. 5e, f).

The CCFv3 labels, and associated ontology were created based on previous works in mice and rats[38,40,41]. Since the nomenclature and abbreviations in same structures are often different between the FP and the CCFv3 labels, we systematically compared between the two labels. For example, cingulate cortex, area 24b (A24b) in the FP labels matches the anterior cingulate area, dorsal part (ACAd) in the CCFv3 labels (Fig. 6i–l). Moreover, the primary motor cortex is abbreviated as M1 in the FP and MOp in the CCFv3 labels (Fig. 6i–l) and the bed nucleus of stria terminalis as ST in the FP and BST in the CCFv3 labels (Fig. 6m–p). We included the complete list of comparisons between the two labels, unique brain region IDs, and hierarchical arrangement in Supplementary Data 2. This information can be

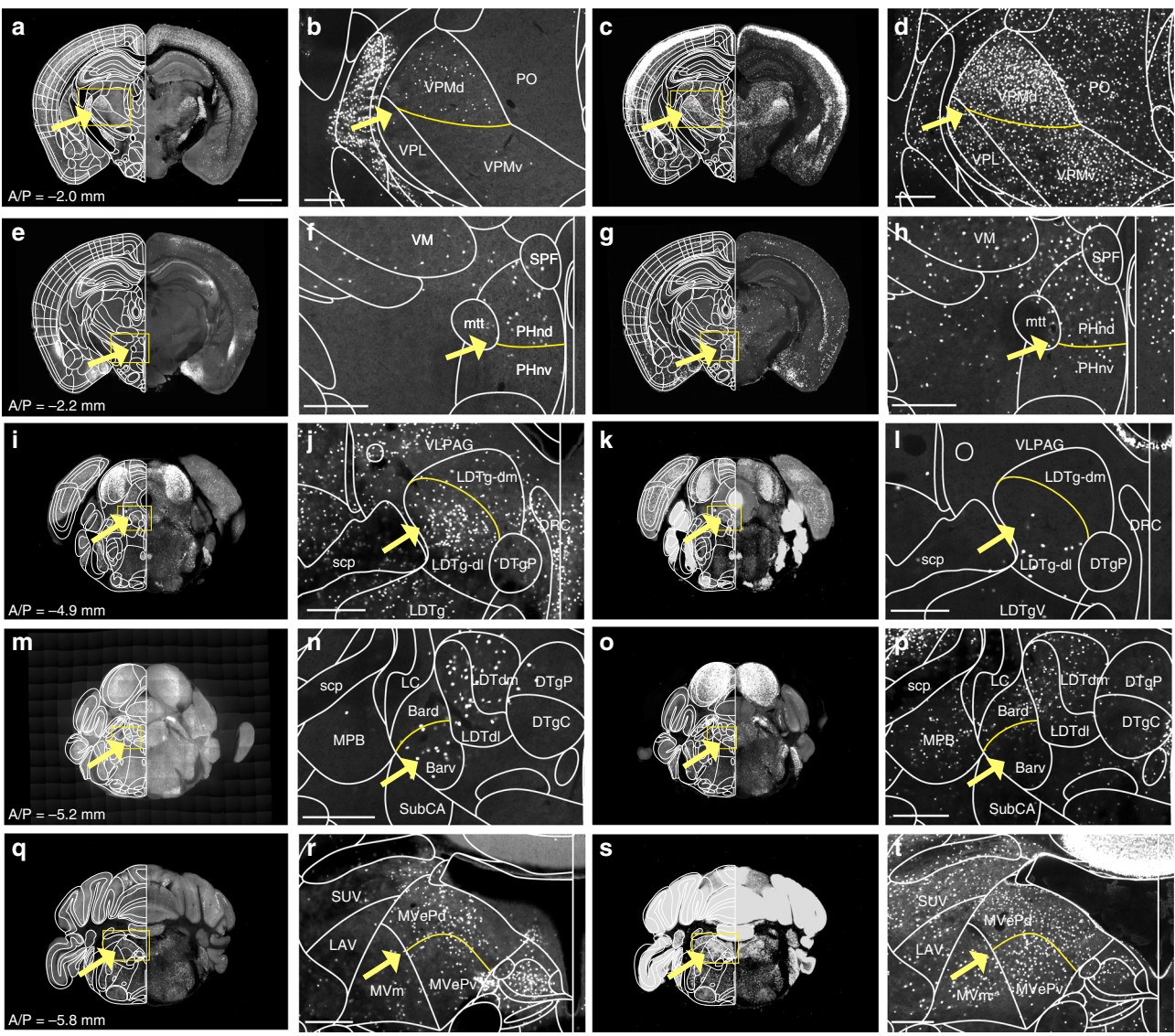

**Fig. 3** Additional segmentations based on marker brains. **a–t** Examples of marker brains to further segment structures defined as a single structure in the FP label. Added segmentations are marked by yellow lines. **a–d** PV-Cre:H2B-GFP (**a**, **b**) and Cux2-Cre:Ai75 (**c**, **d**) marker brains were utilized to further segment ventral posteromedial nucleus of the thalamus (VPM) to dorsal and ventral parts (VPMd and VPMv, respectively). **e–h** OTR-Cre:Ai14 (**e**, **f**) and Ctgf-Cre:Ai75 (**g**, **h**) used to segment dorsal and ventral parts (PHnd and PHnv, respectively) of the posterior hypothalamic nucleus (PHn). **i–l** SST-Cre: H2B-GFP (**i**, **j**) and PV-Cre:H2B-GFP (**k**, **l**) used to segment laterodorsal tegmental nucleus, dorsal part (LDTg) into lateral and medial divisions (LDTg-dl and LDTg-dm, respectively). **m–p** Chat-Cre:Ai75 (**m**, **n**) and SST-Cre:H2B-GFP (**o–p**) used to segment Barrington nucleus (Bar) into dorsal and ventral parts (Bard and Barv, respectively). **q–t** SST-Cre:H2B-GFP (**q**, **r**) and PV-Cre:H2B-GFP (**s**, **t**) used to segment the medial vestibular nucleus, parvicellular part (MVp) to dorsal and ventral parts (MVpd and MVpv, respectively). See Supplementary Data 2 for full names of abbreviations. Scale bars in first and third columns = 2 mm, second and fourth columns = 300 μm

utilized to compare the nomenclature within any brain region between the two atlases.

More detailed workflow of our atlasing work is summarized in Supplementary Fig. 6.

**Comparison between Allen and our FP based anatomical labels.** Because our anatomical labels adopted from the FP labels were aligned in the Allen CCF, we can compare and contrast differences between the two most commonly used anatomical labels in the same space (Fig. 6). We also included the ARA labels drawn in Nissl stained sections as additional comparison (last column of Fig. 6). Our labels have overall finer segmentations than the CCFv3 labels. For example, the zona incerta (ZI) is a part of the subthalamic nucleus that plays an important role in

behaviors such as pain processing and defensive behavior[42,43]. We previously found that parvalbumin (PV) neurons are heavily enriched in ventral ZI[15]. Our FP-based labels segmented PV enriched ventral ZI separately from dorsal ZI while both the ARA and the CCFv3 labels have only one segmentation for ZI (Fig. 6a–d). Moreover, CCFv3 and FP labels often use different boundaries even in similar brain regions. For example, the substantia innominata (SI) in the CCFv3 labels is a part of the basal forebrain structure that is important in attention and learning[44,45]. In our FP-based labels, the matching region is composed of the ventral pallidum (VP), the substantia innominata basal (SIB), and the extended amygdala (EA). In our marker brains, the VP and the EA are marked by cholinergic and somatostatin neurons, respectively (Fig. 6f)[28]. Moreover, a large portion of the EA was included as a part of the lateral preoptic

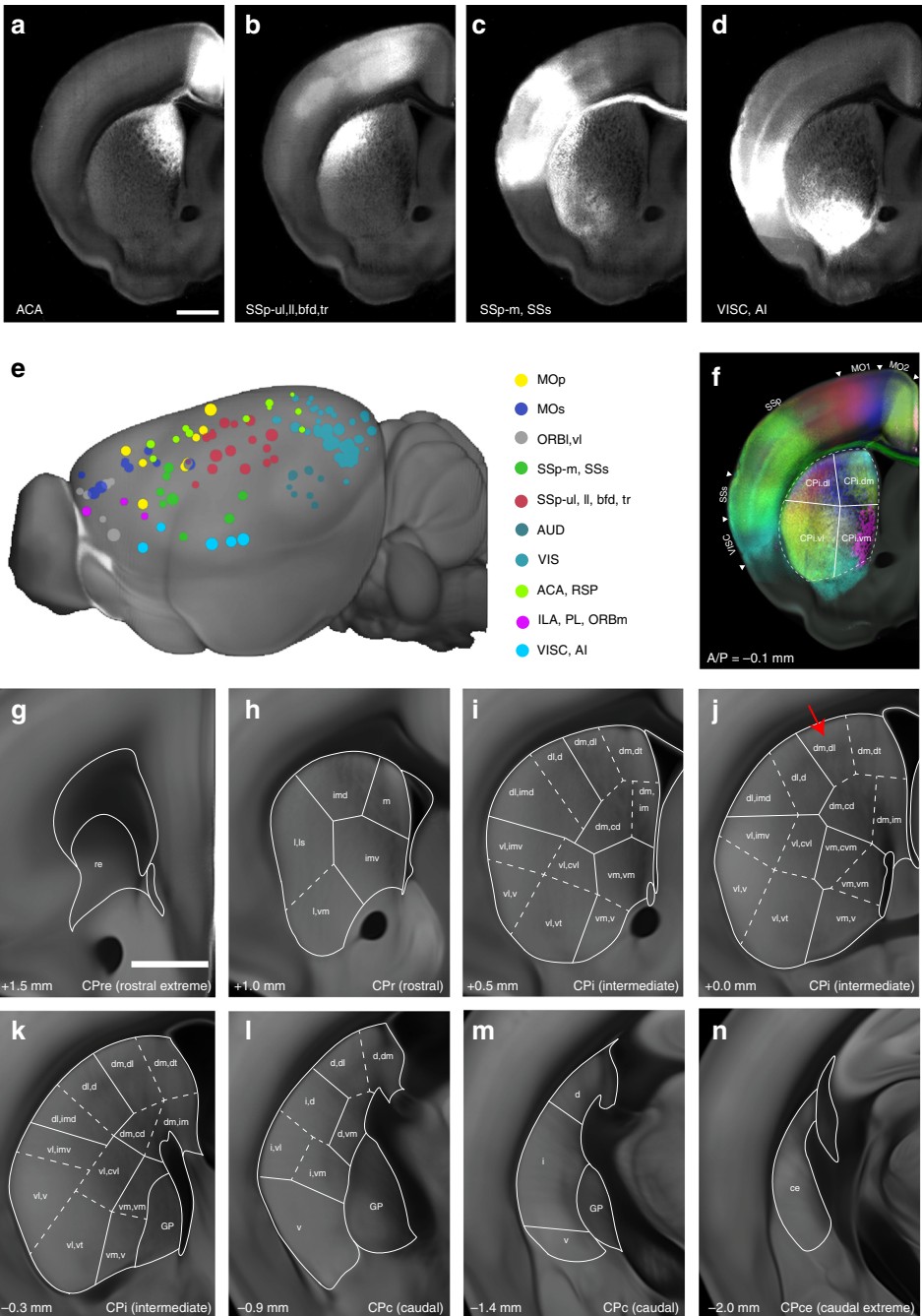

**Fig. 4** Cortico-striatal projection based striatum segmentations. **a–d** Anterograde tracing datasets from different cortical domains registered into the Allen CCF. Scale bar = 1 mm. **a** for the anterior cingulate cortex (ACA), **b** for the primary somatosensory cortex (SSp), upper limb (ul), lower limb (ll), barrel field (bfd), and trunk (tr) area. **c** for the SSp, mouth (m) and secondary (s). **d** for the visceral (VISC) and the agranular insular cortex (AI). **e** 129 datasets clustered into 10 groups based on cortical input regions. Datasets in the same cluster have the same color. **f** Example of CP segmentation based on cortico-striatal projection patterns in CPi regions with 4 different community level segmentations. **g–n** Representative images of CP segmentations throughout several Bregma A/P (number in the left bottom of each figure) planes. imd: intermediate dorsal, m: medial, imv: intermediate ventral, vm: ventral medial, dt: dorsal tip, dl: dorsolateral, dm: dorsomedial, d: dorsal, vl: ventral lateral, v: ventral, cvl: central ventrolateral, cd: central dorsal, im: intermediate dorsal, GP: globus pallidus. Scale bar = 100 µm. Full name of abbreviations can be found in Supplementary Data 2

area (LPO) in the CCFv3 labels (but not in the ARA labels), which does not match with our border between the hypothalamus and the basal forebrain (yellow arrows in Fig. 6f–h). Discrepancies between anatomical borders extend to many different areas including cortical areas. For example, we noticed that the boundary between the motor and the somatosensory cortex in the CCFv3 labels has been dramatically shifted from its ARA label (yellow arrows in Fig. 6j–l). Our labels match better to the ARA

labels than to the CCFv3, consistent with the existence of layer 4 in the somatosensory area, but not in the motor area, and with patterns of cortical layer specific marker brains (Fig. 6j–l). Moreover, the CCFv3 labels simplified segmentation in some key regions that are functionally subdivided. For example, the bed nucleus of the stria terminalis (BST) in the ARA labels was divided into different subregions, but is no longer subdivided in the CCFv3 labels (Fig. 6o, p). BST subdivisions play important

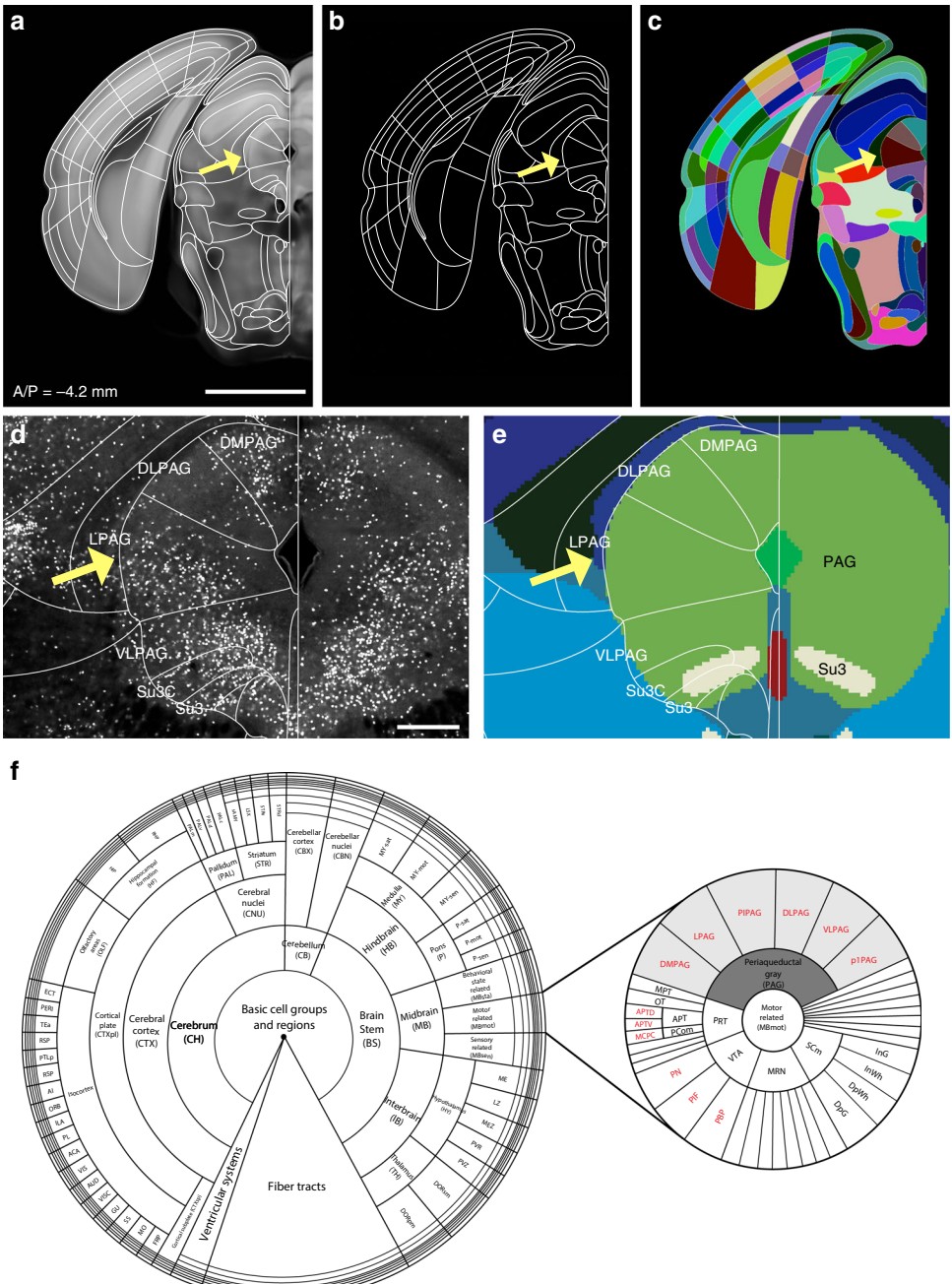

**Fig. 5** Digitization of anatomical structures. **a** Example of our highly segmented FP based labels on the Allen CCF. Yellow arrows highlight the lateral subdivision of periaqueductal gray (PAG), Scale bar = 2 mm. **b** Exported delineation lines. **c** Digitization of labels with unique numerical ID for each anatomical structure. Different color of each structure pertains to different number. **d** SST-Cre:H2B-GFP showed distinct subregions in the PAG with different cell density level. Our labels (white font) divide the PAG into different subregions, as can be seen with the specific enrichment of SST neurons in the dorsolateral PAG (DLPAG) and the lateral PAG (LPAG, yellow arrow). Scale bar = 300 μm. **e** In contrast, the CCFv3 labels (color labels in the background) showed only 2 segmentations within the PAG (black font). **f** Hierarchical organization of anatomical labels based on the Allen ontology. Numerical IDs of individual structures assigned within parent structures for region-level and individual structure-level data analysis. For example, the PAG (shaded dark gray) is the parent structure of six subdivided structures (shaded light gray). Red font labels refer to structures further divided by the FP labels that are not present in the CCFv3 labels. Full name of abbreviations can be found in Supplementary Data 2

roles in distinctive behaviors (e.g., anxiety and social behavior) and have unique anatomical connections[46,47]. Our labels are highly segmented in the BST (Fig. 6m–p).

We extended our analysis to the remaining brain regions and measured the degree of overlap between the three atlases at different ontological levels (Supplementary Fig. 7, Supplementary Data 3). We performed Dice Similarity Coefficient (DSC) analysis to compute the degree of overlap between any given two atlases

(1 = perfect overlap, 0 = no overlap). This approach provides a measure of discrepancies between the atlases. Higher order structures showed overall good overlap (DSC over 0.8) while lower order structures showed pronounced discrepancies (DSC <0.5). For instance, DSC between CCFv3 and our FP based labels is 0.97 in the isocortex, 0.74 in the motor cortex, and 0.49 in the primary motor cortex. In contrast, overlap between the ARA and our labels is 0.95 in the isocortex, 0.80 in the motor cortex, and

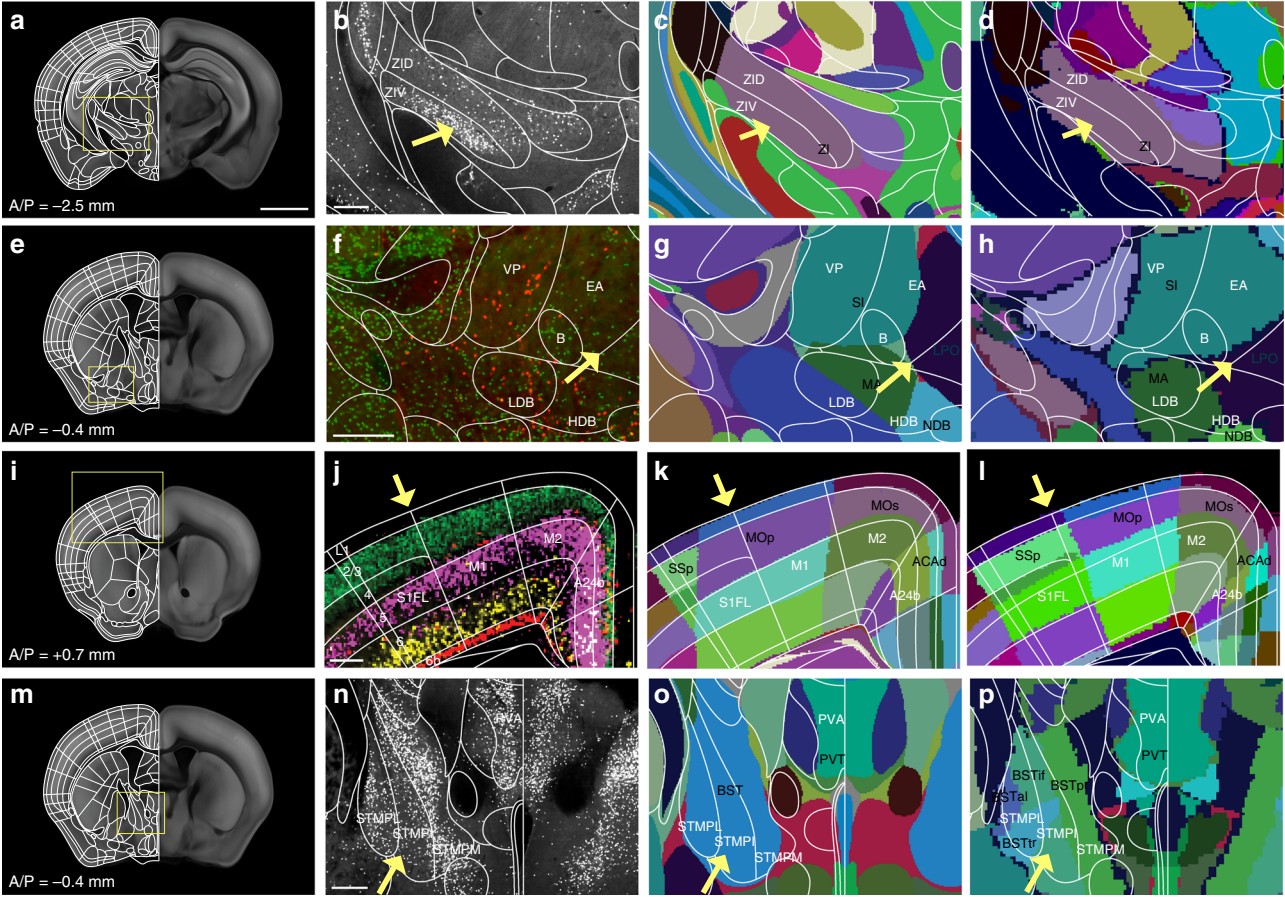

**Fig. 6** Comparison between the Allen CCFv3, the ARA, and our labels. First column: Our highly segmented FP based labels on the Allen CCF. Scale bar = 2 mm, second column; our labels (white lines) with marker brain background, third column: comparison between our labels and the CCFv3 labels (colored background), fourth column: comparisons between our labels and the ARA labels (colored background). **b–p** Anatomical names in black and white are from the Allen CCF and our labels, respectively. **b–d** PV-Cre:H2B-GFP (**b**) to identify subregions in the zona incerta (ZI). Scale bar = 300 μm. Low in dorsal and high in ventral parts (ZID and ZIV, respectively) in our labels while the CCFv3 and the ARA labels have a single combined structure for ZI. **f–h** (**f**) Virtual overlay of Chat-Cre:Ai75 (red) and SST-Cre:H2B-GFP (green) to compare basal forebrain regions. Scale bar = 300 μm. **g**, **h** Our labels further segregate the single structure defined as the substantia innominate (SI, Allen) into the ventral pallidum (VP) and the extended amygdala (EA). Yellow arrow highlights the border between the basal forebrain and the hypothalamus. **j–l** Disagreeing borders between the somatosensory and the motor cortices. Yellow arrow highlights border between the somatosensory and motor cortices. **j** Virtual overlay of pseudo colored Cux2:Ai75 (L2/3, green), Rbp4:Ai75 (L5, magenta), Ntsr1:Ai75 (L6, yellow), and Ctgf:Ai75 (L6b, red). Scale bar = 200 μm. Note the lack of Cux2:Ai75 and Rbp4:Ai75 signal in layer 4 of the somatosensory cortex. **n–p** The BST is divided into several subregions in our labels compared to a single BST structure in the CCFv3 labels, despite the original ARA version with finer delineations for this structure. **n** SST-Cre:H2B-GFP used to identify subdivisions of BST. Scale bar = 300 μm. Note the higher degree of segmentations in our labels compared to the CCFv3 and the ARA (**a–d, e–h, m–p**), and inconsistencies in anatomical delineations of the same structures between the atlases (**i–l**). See Supplementary Data 2 for abbreviations

0.69 in the primary motor cortex (Supplementary Fig. 7). A complete set of comparisons is provided in Supplementary Data 3.

**Web-based atlas visualization and resource sharing**. Web-visualization platforms for digital atlases enable easy identification of anatomical labels across different sections and comparison across different atlases[3,19]. Thus, we created a website (http://kimlab.io/brain-map/atlas/) to visualize and share our anatomical labels. The web visualization includes easy identification of anatomical labels in the background CCF. We recommend using Chrome as the browser to navigate our website. All vector drawing files, digitized labels, and associated files are freely available for download (Supplementary Data 4, 5, and 6). This open source data sharing will facilitate further refinement of anatomical labels and integration of data interpretation within a single anatomical platform.

## Discussion

Here, we present highly segmented open source anatomical labels on the Allen CCF, which are easily accessible via our website. Our labels are largely based on FP labels with cortico-striatal projection based detailed segmentations in the dorsal striatum and further segmentations based on fluorescent transgenic markers.

A reference atlas serves a critical role in understanding the spatial context of the brain[8,11,15,48]. For the mouse brain, a collection of atlases has been generated based on different neuroimaging methods (e.g., MRI) or histological staining[6,7,9–11,25]. For example, Nissl stained 2D sections utilize rich cytoarchitectural information for neuroanatomists to finely delineate anatomical regions[6,7]. In contrast, atlases based on neuroimaging data provide a 3D perspective of the brain with relatively lower resolution and simpler segmentations compared to histology based atlases[11,18]. These atlases exist either in printed or online form (e.g., Waxholm space). However, independently generated atlases with different nomenclature and boundaries can make it difficult

to integrate data from different studies[19]. Significant effort has been made to standardize a rodent brain atlas as a key neuroinformatics tool to facilitate data exchange and to enhance reproducibility between different studies[3,10,19,49]. For example, the International Neuroinformatics Coordination Facility established digital atlas infrastructure for a common spatial framework such as the scalable brain atlas under FAIR (Findable Accessible Interoperable Reproducible) principles[10,19]. Recently, the Allen CCF, generated from iterative averaging of over 1000 different mouse brain samples, provides the highest resolution 3D digital atlas platform[14]. There has been significant encouragement by funding agencies (e.g., BRAIN initiative) to use the CCF as a common anatomical framework for functional and anatomical studies to facilitate seamless exchange between results from different studies[13]. To further support this trend, computational tools are being developed to integrate individual datasets (e.g., 3D imaging or even 2D histological sections) in the standard atlas framework[49–54]. While the CCF provides an ideal atlas platform with high-resolution 3D images, its associated anatomical labels CCFv3 released in 2017 have been controversial due to fewer fine segmentations and significant changes in their anatomical borders from the original version. Moreover, inconsistencies in borders and nomenclature compared to the widely used FP labels make it difficult to compare findings from studies that use different atlases. Discrepancies between atlases can arise from variations in anatomical delineations by different neuroanatomists, different sample preparation, strains, and normal phenotypic variation. For instance, the ARA was based on a fresh-frozen brain while the FP atlas was based on a brain fixed with 4% paraformaldehyde[6,7]. While co-registration of Nissl stained sections from two different atlases may help to reduce the differences, different z sampling rates (e.g., 100 μm for the ARA and 120 μm for the FP atlas) and challenges caused by registering images from different signal content (e.g., Nissl vs autofluorescent background in the Allen CCF) make it difficult to merge two independent atlases in one spatial context. We resolved the issues by combining manual alignments and co-registration of MRI and marker brains.

Our strategy was to establish the FP based anatomical labels in the Allen CCF. We used a series of steps to rigorously align the FP labels in the Allen CCF. We further generated finer segmentations based on marker brains that highlight specific anatomical regions otherwise not visible in the background[22]. These strategies enabled us to establish highly detailed FP based labels in the Allen CCF, which provided a unique opportunity to compare two atlases in the same spatial context. Our comparison revealed substantial discrepancies in anatomical demarcation of same brain regions. Our systematic comparison between the two atlases marks an important first step towards unified anatomical labels in a common atlas platform. As neuroscience research becomes increasingly collaborative, it is essential to have consistency in anatomical labels to specify regions of interest. By integrating FP based labels in the CCF, our labels can be used to facilitate the comparison of anatomical interpretations from past and future studies regardless of the atlas used.

We also used cortico-striatal long-range connectivity to finely segment the dorsal striatum. Projectome-based atlasing provides an alternative way to segment brain regions that do not have distinct cytoarchitectonic features. Since brain-wide projectome data are becoming increasingly available in open source platforms[24,55–57], similar approaches can be used to segment other brain regions with distinct projection patterns. Moreover, since this anatomical connectivity is related to functional interactions between neural circuitry, connectivity based anatomical segmentation can provide a unique opportunity to integrate functional circuits in anatomical maps.

We digitized anatomical labels from vector drawings and organized our labels hierarchically according to the Allen ontology[38,40,41] as a neuroinformatics tool. Thus, our labels can be easily integrated into data processing pipelines to automatically quantify target signals throughout anatomical regions in the whole brain. We previously built such a pipeline to quantitatively map neural activity based on c-Fos induction, GABAergic cell subtypes, and long-range neural connectivity[15,23,56]. Moreover, mapping pipelines are increasingly available for high-resolution 3D image data and histological sections[16,49,58,59]. With image registration to the Allen CCF, our digitized labels can serve as an invaluable neuroinformatics tool to examine target signals in the FP based labels as well as the built-in CCFv3 labels.

Moving forward, by integrating the two most popular brain segmentations in the same 3D anatomical context, our atlas will help to build unified anatomical labels for the mouse brain[3,19,60]. Resolving discrepancies observed between atlases is of growing importance to the future of neuroscience due to constantly changing parameters of neuroimaging and spatially resolved functional characterization. Rapid progress in large scale cell-type and connectivity mapping as well as in vivo neural recording will provide valuable information to refine anatomical borders, especially in areas with less clear cytoarchitectonic features, towards an increasingly rich, unified atlas of the mouse brain. The presented segmentations contain potential errors in boundaries of fine structures due to limitations of currently available marker brains and connectivity datasets. On-going and future efforts to generate large scale cell-type and connectivity database will no doubt facilitate the refining and even further segmentation of anatomical labels[5,13,57]. Moreover, potential sex differences in marker expression may exist, and need to be considered in future work. To facilitate such work, we are making all the data (including vector drawing for delineations) freely available to visualize and download via public data repository and our website. In addition, atlas template brains based on different imaging modalities (e.g., MRI) co-registered to the Allen CCF space will facilitate incorporation of neuroimaging data from the different imaging modalities. Such development can bring us closer to the ideal of allowing data from different imaging modalities to be seamlessly registered into a common atlas space in which detailed anatomical segmentations are available. We envision that similar approaches can be taken to integrate independently generated atlases within animal species including humans.

## Methods

**Animals**. We have complied with all relevant ethical regulations for animal testing and research. All animal work has been approved by the Institutional Animal Care and Use Committee of Penn State University College of Medicine. We used the following transgenic mice to fluorescently label specific cell types (marker brains). For Cre drivers, we used OT-Cre (Jax: 024234), Avptm-Cre (Jax: 023530), and OTR-Cre (gift from Nishimori lab, Tohoku University, not publically available). For Cre dependent reporter mice, we used Ai14 (Jax:007908). We crossed cell type specific Cre driver mice with Ai14 to create maker brains. We used both male and female mice at ~2–3 months old. All mice were group housed in 12/12 light/dark cycle (6 a.m. light on, 6 p.m. off) with access to food and water ad libitum. Other marker brains were downloaded from either publically available BICCN datasets or previously published databases as specified in Supplementary Data 1[15]. Because we observed highly stereotypical expression in each marker brain, we used one representative brain per each marker line for our anatomical work. The complete list of the maker brain with their source is listed in Supplementary Data 1.

**Sample preparation and imaging of transgenic mice**. Transgenic mice were perfused using cardiac perfusion with 0.1 M phosphate buffer (PB) followed by 4% paraformaldehyde (PFA). Brains were post-fixed with 4% PFA at 4 °C overnight and transferred to 0.05 M PB until imaging. The fixed brain was embedded in oxidized 4% agarose and cross linked by 0.05 M sodium borohydride at 4 °C overnight. Oxidized 4% agarose was made by stirring 7 g agarose and 0.735 g sodium periodate in 350 mL of 0.05 M PB for 2.5 h at room temperature in a light-protected beaker under the fume hood. The agarose was filtered three times with distilled water followed by one wash with 0.05 M PB. The agarose was

re-suspended in 175 mL of 0.05 M PB and was stored in 4 °C for brain embedding. In all, 0.05 M sodium borate buffer solution was made with 19 g Borax and 3 g of boric acid in 1 L of distilled water. With 100 mL of this sodium borate buffer solution, the cross-linking solution sodium borohydride was made by adding 0.2 g of sodium borohydrate at room temperature in a light-protected beaker under the fume hood. This can be stored at room temperature for 7 days. For brain embedding, oxidized agarose was heated in the microwave and poured into a custom-built mold with centered brain sample when the agarose cooled down to 65 °C. After solidification at room temperature, the block was trimmed and placed in 50 mL Falcon tube with sodium borhydride for cross-linking at 4 °C overnight. The block was glued to a glass slide with magnetic bars underneath, which was placed in a special chamber with 900 mL of 0.05 M PB for STPT imaging. We used Tissuecyte 1000 (Tissuevision) to perform serial two-photon tomography imaging. We used a 970 nm wavelength laser and acquired a series of images ($12 \times 16$ XY tiles, $700 \times 700$ pixels field of view) at 1 μm X-Y resolution in every 50 μm z section[15]. We used custom-built algorithms to reconstruct the whole brain. Our imaged brains and downloaded marker brains were registered to the Allen CCF based on mutual information using the open source program (Elastix)[61] based on affine and bspline parameters (Supplementary Data 7)[15]. Image registration was performed using 3D image stacks at 20 μm × 20 μm × 50 μm (x,y,z) pixel spacing. Previously, we assessed the warping accuracy using 13 unique 3D landmarks acquired from Waxholm space[10] in 6 different mouse brains after warping each dataset onto the reference brain[23]. Warping accuracy was 65.0 ± 39.9 μm after 3D registration[23], supporting the accuracy of our registration results.

**Importing and modifying the FP labels to the Allen CCF**. We originally obtained vector drawings of Nissl 2D section from Paxinos and Franklin's the Mouse Brain in Stereotaxic Coordinates, 3rd edition[6]. We also used the 4th version to incorporate the latest updated labels. We used a vector drawing tool (Adobe Illustrator) for our labeling work. We downloaded the Allen CCF and associated labels from the Allen Institute for Brain Sciences API (http://help.brain-map.org/display/mousebrain/API), and generated coronal slices (10 μm isotropic) using Image-Stacks-Reslices in FIJI (NIH)[62]. This produced 1320 Z coronal slices. Then, we selected one coronal slice in every 10 slices from Z95 to Z1315 using Image-Stacks-Tools-Make Substack in FIJI, generating 123 coronal images with 100 μm z spacing. We chose 100 μm z spacing to facilitate comparisons between the ARA and the CCFv3 because the ARA was created in coronal sections evenly spaced at 100 μm intervals[38]. We identified matching z planes between the FP atlas and the CCF using distinct anatomical landmarks (e.g., fiber track, and ventricles). Anterior-posterior (A/P) Bregma coordinates of z sections were primarily based on ARA[7] while cross-referencing to the FP atlas[6]. To aid our label alignment in 3D, we downloaded MRI labels from different brain regions from a publically available database (https://imaging.org.au/AMBMC/AMBMC). We combined labels from different brain regions to reconstruct the MRI labels using FIJI (NIH)[62]. Then, we registered the MRI atlas with the FP based labels to the CCF using Elastix. The MRI labels were particularly useful to align boundaries in cortical areas in 3D. We loaded cell type specific labeling from different transgenic mice and MRI labels as separate layers on the Illustrator, and used the information to further adjust anatomical delineations. To accommodate the FP labels (mostly 120 μm z spacing) in 100 μm z spacing, we used the 5th section of every 6 FP labels twice in the initial alignment and used the MRI atlas and marker brains to further modify the labels across the 3D plane. Once the FP labels were imported in the matching plane of the CCF on Adobe Illustrator, we used linear translation to stretch the FP labels to fit the CCF roughly. Then, we performed finer alignment manually based on specific landmarks of the brain with distinct contrast (e.g., fiber tracts). We used shade from levels of background autofluorescence and texture from fine myelinated tracks in the Allen CCF to determine anatomical borders. The shade feature was useful for delineating subregions in the isocortex, the hippocampus, the hindbrain, and the cerebellum; the texture feature was useful in the ventral striatum and the medulla. One or two slides before and after in each 2D section were used to ensure the contiguity of 3D labels. In selected areas (e.g., hypothalamus), boundaries were removed entirely and re-drawn based on key features of the CCF and distinct cell populations. In caudal areas, we often used 2–3 different FP planes to create hybrid labels to fit the CCF background as well as cell type specific features of the selected plane. The primary alignment of each label was performed by U.C., followed by a second and independent inspection by Y.K.

**Connectivity based segmentation in the caudate putamen**. We downloaded 129 datasets with anterograde virus injection in different cortical areas from C57bl/6 mouse line using Allen connectivity database (http://help.brain-map.org/display/mouseconnectivity/API). All downloaded datasets were registered to our modified CCF with 100 um z spacing using Elastix. After image registration, we removed the autofluorescent background of each sample using binary thresholding (FIJI). We clustered projection datasets into 10 groups based on their cortical injection sites and averaged projection signals in the same group using FIJI. Then, we imported the projection data into Illustrator as separate layers and used them to further segment the CP. To import segmentation dataset from Mouse Connectome Project (Center for Integrative Connectomics, University of Southern California), we used

the cortico-striatal map (http://www.mouseconnectome.org/CorticalMap/page/map/5) to add different projection data to the ARA. This website provides a way to overlay up to 10 different projection data sets in 12 different anterior/posterior positions. A set of cortico-striatal projection to distinct CP subregions was selected based on clustering data from Hintiryan et al.[35], and was overlaid in the Allen CCF background for segmentations using Adobe Illustrator. Five different anterior/poster CP levels were determined by overlap, separation, and lack of projection from specific cortico-striatal projections[35,36]. CPi,dm,dl and CPi,dm,d were combined with CPi,dm,dt due to a high degree of in their projection patterns[35].

**Digitization of anatomical labels**. Our labels were first compared to segmented regions of the CCFv3 labels. We used ontologically arranged Allen label numbering system as a template to digitize our labels (Supplementary Data 2). All labels were imported onto FIJI and each region was selected using wand tool and assigned specific anatomical identification numbers using the Process-Math-Add function. If our labels matched the CCFv3 labels, we assigned the same Allen anatomical identification numbers. If our labels were not found in the CCFv3 labels (e.g., finer segmentation in our labels), we assigned unique identification numbers. If there was significant disagreement on the border delineation of matching structures with similar nomenclature, we maintained the same ID number for that specific structure.

**Overlap calculation between atlases**. We used our FP based labels, the CCFv3, and the ARA (each at 100 μm z spacing) to calculate overlap between atlases. The Dice Similarity Coefficient (DSC) was computed for each individual anatomical region including areas in different hierarchical ontology orders across atlases. DSCs were derived from individual absolute volumes (a region in Atlas A and a matched region in Atlas B) as well as the overlap of these volumes as Boolean data using overlap as true positive (TP; overlap), false positive (FP; A but B), and false negative (FN; B but A) between each pair of atlases as listed in Supplementary Data 3. $DSC = 2TP (2TP + FP + FN)^{-1}$.

**Reporting summary**. Further information on research design is available in the Nature Research Reporting Summary linked to this article.

## Data availability
Data that support the findings of this study and new data from the current study are available in Dryad data (https://doi.org/10.5061/dryad.t1g1jwsxw). Additional data are available from the authors on reasonable request. Following data are obtained from publically available sources. MRI labels (https://imaging.org.au/AMBMC/AMBMC): hippocampus, cerebellum, cortex, basal ganglia, diencephalon labels. Allen Connectivity dataset (http://help.brain-map.org/display/mouseconnectivity/API): viral tracing dataset from isocortical areas from C57bl/6 mice. Mouse Connectome Project (http://www.mouseconnectome.org/CorticalMap/page/map/5): cortico-striatal projection map. BICCN cell type data (http://www.mouseconnectome.org/CorticalMap/page/map/5): Chat_Ai75_M_382462, Emx1_Ai75_M_343525, Gad2_Ai75_M_398912, Ctgf-T2A_Ai75_M_395411, Ntsr1_Ai75_M_369820, Rbp4_Ai75_M_392433, Cux2_Ai75_M_384010.

## Code availability
Custom built stitching algorithm to reconstruct images from serial two-photon tomography and Elastix for image registration were publicly distributed in Kim et al.[15]. Elastix registration parameter files can be found in Supplementary Data 7. Our python based code to perform Dice Similarity Coefficient calculation can be found in the Dryad data (https://doi.org/10.5061/dryad.t1g1jwsxw) under "6_Atlas-comaparison_Dice". All codes can be used without any restriction.

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

## Acknowledgements

We thank Rhea Sullivan in helping generating cortico-striatal projectome data, and Pavel Osten and Piotr Majka for critical reading and editing the manuscript. This publication was made possible by NIH grants (R01MH116176, R01NS108407) and Tobacco Cure Funds from the Pennsylvania Department of Health to Y.K. and facilitated by NIH grant R24OD018559-01-A2 to K.C. Its contents are solely the responsibility of the authors and do not necessarily represent the views of the funding agency.

## Author contributions

Conceptualization by Y.K.; label alignment and digitization by U.C.; Dorsal striatum segmentation by Y.K.; Web visualization and DSC calculation by D.J.V.; Manuscript preparation by Y.K., U.C., and K.C.C.

## Competing interests

The authors declare no competing interests.
