## [Transparent Peer Review File · Nature Communications]

Reviewers' comments:

Reviewer #1 (Remarks to the Author):

The authors present in this manuscript a computational integration of the Franklin-Paxinos mouse brain atlas (FP) and the Allen reference mouse brain atlas (CCF) to generate a more comprehensive atlas for general use in neuroscience. The authors in addition use data from 14 transgenic mouse lines to label the distribution of some neuron subtypes, as well as connectivity data for dorsal striatum to provide new definitions of the striatal volume in this new reference atlas. This is overall an ambitious effort to improve on the available mouse brain mapping resources, and is in line with the community efforts to generate better definitions of mouse brain subregions.

Major issues

It is unclear from the current manuscript how this work is different from a previous publication in Cell (Kim Y, et al, 2017). The authors should be clear on this issue.

From the figures, this seems to be a FP coordinate system, but the authors should explain this aspect since it is of fundamental importance when neuroscientists use coordinates to describe position in the mouse brain.

It is unclear if the authors have generated a new 3D atlas using their new annotations, and how this space then is represented and visualized (e.g. polygons, NURBS). The authors should provide their atlas as a 3D resource, with clear instructions how to download the vector graphics in order for other labs to use and modify this atlas. This is important if the authors wish to have an impact on how the field is currently using reference maps.

Many of the markers selected by the authors are classical markers for hypothalamic populations. The authors should use this information to update the anatomy of the hypothalamus (HY), where annotation of borders in CCF and FP is probably inaccurate and inconsistent. A figure similar to Figure 3, but focusing on HY, would be very valuable to the field.

In Figure 3, the new segmentation of thalamic nuclei should be compared to connectivity data from the Allen Institute (Oh SW, et al, Nature 2014). The authors should also discuss the work in relation to the work on definitions of thalamic nuclei based on gene expression (e.g. Nagalski A, et al, Brain Struct Funct 2016; Phillips JW, et al, Biorxiv 2018).

In Figure 5, PAG subdivisions are made using Sst-Cre:H2B-GFP: what about connectivity data from Allen Institute? Here the authors should show some connectivity examples that support these PAG subdivisions. Otherwise these subdivisions are arbitrary and not supported by data.

The authors have added 10 new subdivisions based on the marker expression, but the information on these is presented in text format in Table S2. As was done for Figure 3, the authors should provide high-resolution supplementary images showing the new nuclei/border definitions for all new subregions. These images should include information on the unique identification number of these subregions and show the border differences comparing to both FP and CCF.

Minor issues

What were the selection process for the transgenic mouse lines? – the authors should discuss the limitations of this approach since it is a biased selection of markers.

The online resource is a valuable tool, and looks in design very similar to the published atlas openbrainmap.org, and the authors do not cite this publication (Furth D, et al, Nature Neuroscience 2018). The authors should also consider citing key publications for the software(s) they use, for example FIJI.

Better labels in figures and description in the figure legends should be improved to help readers to easily understand the data in the different panels. As is the case now, it is difficult to follow what atlas is used in different panels and what message each comparison is supposed to convey.

Very little information on the supplementary files, these need to be annotated and explained in a Supplementary text.

The supplementary figures are not color-coded according to the CFC scheme, are only greyscale in the version I downloaded.

In the methods section and the Table S1, the authors describe the use of “Avptm-Cre mice” (Avp-Cre mice to label vasopressin+ cells), but no data is found in the figures about the results of this. Please update this and include figure of marker distribution and how this is used in new definitions of subregions.

It is unclear if sex difference in marker expression was taken into account or analyzed, but this should be discussed.

The authors should also extend their discussion with more in-depth examples of the discrepancies and controversies found in the field of mouse brain maps.

There are formatting errors in the last pages in the PDF version of Table S1.
There are formatting errors in the last pages in the PDF version of Table S2.

Reviewer #2 (Remarks to the Author):

Chon et al. describe in the paper “Enhanced and unified anatomical labeling for a common mouse brain atlas” their approach to merge the brain labels from the classical Paxinos & Franklin mouse brain atlas and the more recent version of the Allen Mouse Brain Reference atlas (ccf v3). They merged labels, updated and created new labels using cell type specific transgenic mice and the AMBMC MRI atlas. The resulting atlas is available through an interactive web interface. While I appreciate the authors approach to create an improved and common atlas, I have concerns regarding the methodology and the manuscript. Especially the critical process of 3-D annotation is not well described and the criteria for the anatomical definition are missing. For example: How did the authors decide which pixels belong to a certain brain area; using pre-defined criteria, verification by multiple expert raters, best guess? The atlas is available online, however, the interface is useful for viewing only. There are no further options, even no simple search function. The atlas plates contain only half side labels and border zones between the labels which makes it not useful for further use, especially image registration (the border zones = 0 would be stretched/tilted).

In order to convince people to switch from the other atlases to this novel atlas, the methodology needs to be re-written, more convincing improvements described (and visualized) and important functions (such as the search function) added to the website.

Major comments:

Introduction

- Page 1, line 5: evidence for that statement is missing, authors neglect that there are actually a variety of atlases available to data, see lists provided for example in Hjernevik et al. 2007 (<https://www.frontiersin.org/articles/10.3389/neuro.11.004.2007/full>) and Pallast et al. 2019 (<https://www.frontiersin.org/articles/10.3389/fninf.2019.00042/full#supplementary-material>). The same is true for the discussion, where not only other atlases should be compared in more detail but also other atlas spaces (e.g. the Waxholm space).
- Page 1, line 21 ff:). the ARA ontology was adopted from Swanson et al. 2004 and Hof et al. 2000 (see AllenReferenceAtlas_v1_2008 Technical White Paper for details), if it is about the differences in brain region names the authors should discuss these references in more detail and give 2-3 examples (best with representative images)
- Line 26 ff: what is different to the approach described by Hjernevik et al. 2007?
- Line 28 ff: references missing, I actually doubt that; the ARA ccf v3 comes with >1300 structures (including parental and childs)

Material&Methods

- Page 9, line 29 ff: the procedure on how the authors created the 3D labels is not well written and the selection of reference slides (e.g. every 10th ARA) seems to be arbitrary – at least it is not further explained. What puzzles me most: what where the criteria to decide for the CCF or FP/AMBMC landmark/region boundary in a particular situation and who did that, a single person? An overview figure would be helpful to understand the workflow (c.f. page 5, Allen Mouse CCF white paper)
- Page 9, line 44: Registration details missing, rigid/affine etc., make the code available in order to allow other users to register their data sets and further improve the atlas. I think that it is the crucial part of the manuscript as the registration of the atlas plates will induce the largest error if it is not done well - meaning related to specific anatomical landmarks (e.g. through a landmark correspondence approach, see Ito et al. Stroke 2018 <https://www.ahajournals.org/doi/10.1161/STROKEAHA.118.021508>)

Minor comments:

Introduction

- FP and Allen are quite uncommon abbreviations, especially Allen could be mistaken for Paul G. Allen, the founder of the Allen Institute – use ARA instead
- Line 16: the 3D atlas is also particularly useful for in vivo imaging (PET, MRI, etc) see references to my comment on line 5
- Typos: line 9 stainings, 19 delineations, 17 sharing of...I stop counting the errors made with the plural s, there are just too many. Revise the whole manuscript for typos.

Reviewer #3 (Remarks to the Author):

In this manuscript, Chon et al. reports a valuable effort of developing a “translatable” 3D mouse brain atlas. Firstly, the authors registered anatomical labeling of two commonly used mouse brain atlases, Franklin and Paxinos (FP) and Allen Reference Atlas (Allen), into the same 3D Allen Common Coordinate Framework (CCF), allowing users to directly visualize and compare anatomical delineations and nomenclature of the mouse brain in the same 3D space frame. Secondly, they used cell type specific transgenic mice and an MRI atlas to adjust and further refine anatomical delineations. Thirdly, they refined the dorsal striatum (CP) into multiple subdivisions based on cortico-striatal connectivity data of the Allen Connectivity Database. Lastly, they have digitized the anatomical labels based on the Allen Reference Atlas ontology, created a web-interface for visualization, and provided tools for comprehensive comparisons between the Allen and FP labels.

This study is a valuable exploration approach to address one fundamental problem in the field of neuroanatomy: how to address discrepancies among different brain atlases - for example, the FP and Allen, by constructing a unified mouse brain atlas. In the last decade, along with the rapid development of microscopic and computational technologies, digital brain atlases have become an effective informatics tool to facilitate data analysis and visualization, in addition to their classic role in referential texts. The Allen Reference Atlas (Dong, 2008) was the first digital atlas to serve this role and facilitate the development of the Allen Brain Atlas gene mapping project, Allen Connectivity Atlas, and other large-scale brain projects. The current study for the first time converts the FP delineations into a 3D-brain frame thereby allowing it to be directly compared with Allen delineations. This effort will be very well-received by the neuroscience community. Overall, the manuscript is well-written, and all figures are of high quality. Therefore, I strongly support its publication in Nature Communications.

My major comment is about the comparison of the FP and Allen. The FP, Allen CCF, Allen Reference Atlas, and MRI atlas are all in different imaging modalities and at different resolutions. First, it will be very useful for the authors to generate a table to show a detailed comparison of which structures are well-matching (let's say, >80% overlapping) between each of those three atlases (FP, Allen CCF, and Allen Reference Atlas) and how many structures display large discrepancies (i.e., < 50% overlapping?). Second, from my view, some of those discrepancies between the FP and Allen or between the Allen CCF and original Allen Reference Atlas are obviously due to different anatomical delineations from different neuroanatomists. However, some mismatching of labeling might be due to different manners of tissue preparation and imaging process. The FP atlas was constructed using a brain from a mouse with 4% PFA perfusion (which may cause shrinkage of brain tissue); while the original Allen Reference Atlas was constructed with a fresh-frozen brain. Therefore, some mismatching labeling might be due to non-linear deformation of the brain tissues. The authors may want to discuss this issue and potential solution to reduce these discrepancies. For example, whether it will reduce the discrepancy of those two sets of labeling if they can register the corresponding Nissl-stained coronal sections (not just anatomical labeling) of FP and Allen Reference Atlas together. The authors may also want to discuss in more details about other technical challenges and potential solutions of constructing this kind of "unified " brain atlas(es).

My other comments are mainly focused on their delineations of the striatum based on Allen Connectivity data: First of all, a recent report by Hintiryan et al. (Nature Neuroscience, 2016) produced a comprehensive anatomical delineation with fine-detailed nomenclature of the dorsal striatum based on the USC Mouse Connectome Project. It appears that by comparison, these two sets of delineations (the new delineation proposed in this manuscript and the one in Hintiryan et al.) have certain discrepancies. Therefore, to complete the analytical rigor of this report, the author may want to also adopt the delineations of Hintiryan et. al. into the same space frame. I also suggest that the authors consider generating different nomenclature for the new subdivisions of the striatum since the current nomenclature of calling new subdivisions of the striatum with different cortical areas is very unclear. Figure 3 in Hintiryan et. al. (2016) provides an excellent example that can be referenced for the authors' consideration.

The authors should be aware that the vast majority injections of the Allen Connectivity data are relatively large and involved in multiple cortical areas –the projection pattern may not be "clean." Creating parcellations of the CP and labelling them with particular cortical areas such as "ACA" "MOs" may cause confusion. Furthermore, the authors should be aware that each of these subdivisions receive inputs from multiple cortical areas. For example, axonal terminals from ACA and RSP are overlapping in the CP. Again, labeling that part of the CP as the "ACA" may be misleading. Finally, the ORB-vI subdivision does not look fully accurate to me (seems slightly too ventral).

Reviewer #4 (Remarks to the Author):

Enhanced and Unified Anatomical Labeling for a Common Mouse Brain Atlas

The authors present their efforts to integrate and unify the Franklin and Paxinos (FP) mouse brain atlas and the Allen Institute's common coordinate framework (CCF). They provide the result of this integration open and freely available.

The authors have provided an excellent service to the community by working through a number of challenging issues in integrating two widely adopted mouse brain atlases. Using a combination of MRI, cell-type specific labeling and long-range connectivity data they provide the labels of the FP atlas in the CCF space. The result is a valuable contribution to spatial data integration for the neuroscience community.

However, a few items would improve the manuscript:

The discussion focuses more on the motivation (which really is in the introduction). I would recommend that the Discussion highlight the limitations and opportunities for future improvements. The text would do well to have a caveats discussion to recognize the challenges of integrating multiple modalities of data, challenges in ensuring boundaries correspond and estimates of the potential error and which brain region are most vulnerable to mapping errors.

The other issue is the terminology. Occasionally, the Allen Institute for Brain Science Common Coordinate Framework is referred to simply as "the Allen". I would suggest using "the Allen CCF" as a minimum.

Response to the Reviews

Article: Chon et al., “Enhanced and Unified Anatomical Labeling for a Common Mouse Brain Atlas”.

We thank the reviewers for their constructive comments and suggestions. We have addressed the reviewers concerns as detailed below.

Reviewer #1 (Remarks to the Author):

The authors present in this manuscript a computational integration of the Franklin-Paxinos mouse brain atlas (FP) and the Allen reference mouse brain atlas (CCF) to generate a more comprehensive atlas for general use in neuroscience. The authors in addition use data from 14 transgenic mouse lines to label the distribution of some neuron subtypes, as well as connectivity data for dorsal striatum to provide new definitions of the striatal volume in this new reference atlas. This is overall an ambitious effort to improve on the available mouse brain mapping resources, and is in line with the community efforts to generate better definitions of mouse brain subregions.

We thank the reviewer for the enthusiasm of our work.

Major issues

It is unclear from the current manuscript how this work is different from a previous publication in Cell (Kim Y, et al, 2017). The authors should be clear on this issue.

The previous publication (Kim et al., 2017, Cell) focused on quantifying GABAergic neuronal subtypes using **existing** anatomical labels based on Allen Reference Atlas. Major emphasis from Kim et al., 2017 paper was to demonstrate quantitative difference of GABAergic subtypes across different brain regions. During this study, we noticed significant discrepancies between the Allen Reference Atlas and the other popular atlas of Franklin and Paxinos. This observation motivated us to integrate these two atlases into one spatial framework, as presented in the current manuscript.

Importing the Franklin-Paxinos based atlas into the Allen common coordinate framework (CCF) allowed us to unify the two most commonly used sets of anatomical labels in the single spatial context. Thus, our current work provides vetted alternative anatomical labels as a new neuroinformatics tool for interpretation of anatomical regions in the CCF.

From the figures, this seems to be a FP coordinate system, but the authors should explain this aspect since it is of fundamental importance when neuroscientists use coordinates to describe position in the mouse brain.

Anterior/posterior (A/P) coordinates are primarily based on the Allen Reference Brain (ARA). When the ARA was compared to the FP atlas, A/P coordinates mostly matched between the two atlases.

We added following text in the method section of the manuscript.

“Anterior-posterior (A/P) Bregma coordinates of z sections were primarily based on ARA ⁷ while cross-referencing to FP atlas ⁶.”

It is unclear if the authors have generated a new 3D atlas using their new annotations, and how this space then is represented and visualized (e.g. polygons, NURBS). The authors should provide their atlas as a 3D resource, with clear instructions how to download the vector graphics

in order for other labs to use and modify this atlas. This is important if the authors wish to have an impact on how the field is currently using reference maps.

Our new labels are based on vector drawing in coronal planes of CCF at 100 μm z intervals. All vector drawing files with the CCF background are included in the manuscript as supplementary file 3 in this revision. We also provided a series of coronal slices with digitized labels as supplementary file 1. 3D stacks of these 2D labels can serve as a resource to visualize structures in 3D. Furthermore, our updated web visualization can help to navigate our labels across different Z slices.

We added the following text in the result section (page 7).

“All vector drawing files, digitized labels, and associated files are freely available for download (Supplementary File 1, 2, and 3).”

Many of the markers selected by the authors are classical markers for hypothalamic populations. The authors should use this information to update the anatomy of the hypothalamus (HY), where annotation of borders in CFC and FP is probably inaccurate and inconsistent. A figure similar to Figure 3, but focusing on HY, would be very valuable to the field.

Agreed. We indeed found many marker brains that were very useful for delineating hypothalamic nuclei. We added a new figure (Figure S2) describing hypothalamic delineations with marker brains.

In Figure 3, the new segmentation of thalamic nuclei should be compared to connectivity data from the Allen Institute (Oh SW, et al, Nature 2014). The authors should also discuss the work in relation to the work on definitions of thalamic nuclei based on gene expression (e.g. Nagalski A, et al, Brain Struct Funct 2016; Phillips JW, et al, Biorxiv 2018).

Agreed. We have now subdivided the ventral posteromedial nucleus of thalamus (VPM) into dorsal (VPMd) and ventral (VPMv) components based on differential parvalbumin and Cux2 expression. We used the spatial search function in the Allen connectivity database based on Oh SW, et al, Nature 2014. We found differential cortical input to these two subdivisions of the VPM. The VPMd preferentially received input from the anterior somatosensory area (e.g., SSp-mouth) while the VPMv received stronger input from the posterior somatosensory area (e.g., SSp-barrel field).

We created a new figure (Figure S4) and added following text on page 5:

“We further examined whether new VPM subdivisions created here are supported by differential neural connectivity, using the Allen Mouse connectivity database²⁴. Indeed, VPMd and VPMv preferentially received inputs from anterior and posterior cortical area, respectively (Figure S4).”

Moreover, we added following text on page 5 to acknowledge previous studies that used gene expression to differentiate thalamic regions:

“Previous studies have used gene expression patterns to delineate the different thalamic nuclei^{33,34}.”

In Figure 5, PAG subdivisions are made using Sst-Cre:H2B-GFP: what about connectivity data from Allen Institute? Here the authors should show some connectivity examples that support these PAG subdivisions. Otherwise these subdivisions are arbitrary and not supported by data. The spatial search function in Allen connectivity database was used to investigate long-range input to PAG subdivisions. For example, inputs from different brain regions project to distinct subregions of the PAG. We created Figure S5 to support our observation and added the following text on page 6.”

“Anatomical connectivity data also showed that these subdivisions receive topographically distinct inputs from other brain regions (Figure S5).”

The authors have added 10 new subdivisions based on the marker expression, but the information on these is presented in text format in Table S2. As was done for Figure 3, the authors should provide high-resolution supplementary images showing the new nuclei/border definitions for all new subregions. These images should include information on the unique identification number of these subregions and show the border differences comparing to both FP and CCF.

We updated Figure 3 to include high-resolution images of the remaining subdivisions. Furthermore, we created Figure S3 to show unique ID of new subregions and provided comparison between FP and CCF labels.

Minor issues

What were the selection process for the transgenic mouse lines? – the authors should discuss the limitations of this approach since it is a biased selection of markers.

We used a set of 14 mice from different neuropeptides, neurotransmitters, transcription factors, G protein coupled receptors, calcium binding proteins, and a growth factor.

We specified our selection criteria in the results section (page 4):

“We chose marker brains from different neuropeptides, neurotransmitters, transcription factors, G protein coupled receptors, calcium binding proteins, and a growth factor that highlight anatomical boundaries otherwise often not visible in the CCF tissue autofluorescent background (Table S1)”

We also updated Table S1 with an added column (“category”) to specify the gene category for each transgenic marker.

In the discussion section (page 10), we added the following text to acknowledge our current limitation to 14 marker brains.

“The presented segmentations may contain potential errors in boundaries of fine structures due to limitations of currently available marker brains and connectivity datasets.”

The online resource is a valuable tool, and looks in design very similar to the published atlas openbrainmap.org, and the authors do not cite this publication (Furth D, et al, Nature Neuroscience 2018). The authors should also consider citing key publications for the software(s) they use, for example FIJI.

Our apologies for missing this important paper. We added the citation for Furth D, et al, Nature Neuroscience 2018 in our discussion section (page 9 and 10).

Although our web visualization is somewhat similar to the map in openbrainmap.org, we independently developed our web site without knowledge of the openbrainmap website.

We have added citation of software including FIJI in the methods section.

Better labels in figures and description in the figure legends should be improved to help readers to easily understand the data in the different panels. As is the case now, it is difficult to follow what atlas is used in different panels and what message each comparison is supposed to convey.

Our apologies for this confusion. We assigned Allen Reference Atlas (ARA) and Allen CCF to refer the original Allen label and the updated Allen labels, respectively. We applied these

designations throughout the manuscript including figure legends to clearly indicate what atlas is being used.

We also added the following text in the figure 6 legend.

“Note the higher degree of segmentations in our labels compared to the Allen CCF and the ARA (A, B, and D), and inconsistencies in anatomical delineations of the same structures between the atlases (C).”

Very little information on the supplementary files, these need to be annotated and explained in a Supplementary text.

We have added more detailed descriptions to the supplementary files, and added texts to explain the purpose and use of each file in more detail.

The supplementary figures are not color-coded according to the CFC scheme, are only greyscale in the version I downloaded.

We believe that the reviewer refers to supplementary File 1 rather than the supplementary figure.

We updated the Supplementary File 1 with bilateral 16-bit grey scale images that are color coded based on distinct numerical ID, which can be cross-referenced with Table S2. This format can be easily integrated with data processing pipelines to quantify signal in regions of interest.

Our color-coded version of our segmentations can be also found in our web visualization at <http://kimlab.io/brain-map/atlas>.

In the methods section and the Table S1, the authors describe the use of “Avptm-Cre mice” (Avp-Cre mice to label vasopressin+ cells), but no data is found in the figures about the results of this. Please update this and include figure of marker distribution and how this is used in new definitions of subregions.

We have now updated Figure S1 to include delineations associated with Avptm-Cre mice

It is unclear if sex difference in marker expression was taken into account or analyzed, but this should be discussed.

Sex differences in marker expression were not considered in the current study. To acknowledge our limitation, we added the following text in the discussion.

“Moreover, potential sex differences in marker expression may exist, and need to be considered in future work.”

The authors should also extend their discussion with more in-depth examples of the discrepancies and controversies found in the field of mouse brain maps.

We have added the following text to include more in-depth examples of discrepancies and controversies in our discussion:

“For the mouse brain, a collection of atlases has been generated based on different neuroimaging methods (e.g., MRI) or histological staining^{6,7,9-11,25}. For example, Nissl stained 2D sections utilize rich cytoarchitectural information for neuroanatomists to finely delineate anatomical regions^{6,7}. In contrast, atlases based on neuroimaging data provide a 3D perspective of the brain with relatively lower resolution and simpler segmentations compared to histology based atlases^{11,18}. These atlases exist either in printed or online form (e.g., Waxholm space).”

“Discrepancies between atlases can arise from variations in anatomical delineations by different neuroanatomists, different sample preparation, strains, and normal phenotypic variation. For

instance, the ARA was based on a fresh-frozen brain while the FP atlas was based on a brain fixed with 4% paraformaldehyde^{6,7}. While co-registration of Nissl stained sections from two different atlases may help to reduce the differences, different z sampling rates (e.g., 100 µm for the ARA and 120 µm for the FP atlas) and challenges caused by registering images from different signal content (e.g., Nissl vs autofluorescent background in the Allen CCF) make it difficult to merge two independent atlases in one spatial context. We resolved the issues by combining manual alignments and co-registration of MRI and marker brains.”

There are formatting errors in the last pages in the PDF version of Table S1.

There are formatting errors in the last pages in the PDF version of Table S2.

This error occurs in the conversion process of MicroSoft Excel files to pdf format. We encourage the use of MicroSoft Excel to open supplementary table files.

Reviewer #2 (Remarks to the Author):

Chon et al. describe in the paper “Enhanced and unified anatomical labeling for a common mouse brain atlas” their approach to merge the brain labels from the classical Paxinos & Franklin mouse brain atlas and the more recent version of the Allen Mouse Brain Reference atlas (ccf v3). They merged labels, updated and created new labels using cell type specific transgenic mice and the AMBMC MRI atlas. The resulting atlas is available through an interactive web interface.

While I appreciate the authors approach to create an improved and common atlas, I have concerns regarding the methodology and the manuscript.

We thank the reviewer for the enthusiasm and constructive criticisms.

Especially the critical process of 3-D annotation is not well described and the criteria for the anatomical definition are missing. For example: How did the authors decide which pixels belong to a certain brain area; using pre-defined criteria, verification by multiple expert raters, best guess?

We initially used texture and shade features of the background autofluorescence in the Allen CCF to align the original FP labels. Then, we used MRI labels registered to the Allen CCF to guide our alignments in areas without distinct autofluorescent contexts (e.g., borders between cortical regions). MRI labels also provided a reference point of 3D annotation. This procedure is summarized in Figure 1. Then, we used cell type specific marker brains to identify anatomical regions with known cell type marker expression to further refine our labels (Figure 2). The atlas work was initially done by the first author (Uree Chon), followed by independent validation by the senior author (Yongsoo Kim). We have added Figure S6 to explain our entire work flow.

We added the following text in the methods section (page 13).

“We used “shade” from levels of background autofluorescence and “texture” from fine myelinated tracks in the Allen CCF to determine anatomical borders. The shade feature was useful for delineating subregions in the isocortex, the hippocampus, the hindbrain, and the cerebellum; the texture feature was useful in the ventral striatum and the medulla. One or two slides before and after in each 2D section were used to ensure the contiguity of 3D labels.”

“The primary alignment of each label was performed by U.C., followed by a second and independent inspection by Y.K.”

The atlas is available online, however, the interface is useful for viewing only. There are no further options, even no simple search function.

Based on our agreement with this comment, we have made significant improvements to our web interface.

- We added “search function” to identify regions of interest.
- Ontology based label structures have been added.
- “Label” buttons enable the user to highlight regions of interest in the atlas plate
- Label abbreviations have been overlaid in corresponding areas
- We updated our user tutorial video in the landing page, explaining how to use the web visualization tools.

The atlas plates contain only half side labels and border zones between the labels which makes it not useful for further use, especially image registration (the border zones = 0 would be stretched/tilted).

To address these issues, we updated our label files with bilateral sides and removed border zones from Supplementary File 1. We also added Supplementary File 3 for the vector drawing files to make the drawing easily editable for future refinement. For the web visualization, we displayed the label on the left and the corresponding CCF background in the right side to highlight regions of interest in the CCF background. Upon selecting any label on the left side, the corresponding label will appear in the CCF background on the right side. This allows users to identify regions of interest easily in the anatomical context on the CCF.

In order to convince people to switch from the other atlases to this novel atlas, the methodology needs to be re-written, more convincing improvements described (and visualized) and important functions (such as the search function) added to the website.

To address these issues, we have added more detailed descriptions of our atlasing in the methods section as described above.

We added Figure S2 to demonstrate how we used marker brains to align the hypothalamic regions, as additional examples of how alignments were accomplished.

We have also added “search function” and many other visualization tools including global and local anatomical label lists to our website.

Major comments:

Introduction

- Page 1, line 5: *evidence for that statement is missing, authors neglect that there are actually a variety of atlases available to data, see lists provided for example in Hjernevik et al. 2007 (<https://www.frontiersin.org/articles/10.3389/neuro.11.004.2007/full>) and Pallast et al. 2019 (<https://www.frontiersin.org/articles/10.3389/fninf.2019.00042/full#supplementary-material>). The same is true for the discussion, where not only other atlases should be compared in more detail but also other atlas spaces (e.g. the Waxholm space).*

Thank you for pointing out these important papers.

- 1) We added Hjernevik et al. 2007 to the introduction and Pallast et al. 2019 to the discussion.
- 2) To the introduction, we added:
“...the most widely used animal model to understand the mammalian brain, a variety of printed and/or digital atlases exist with varying levels of segmentations in 2D or 3D images acquired from different imaging modalities (e.g., Nissl staining, or MRI) ⁶⁻¹¹”
- 3) To the discussion, we added the following acknowledgment of existing atlases:
“For the mouse brain, a collection of atlases has been generated based on different neuroimaging methods (e.g., MRI) or histological staining ^{6,7,9-11,25}. For example, Nissl stained 2D sections utilize rich cytoarchitectural information for neuroanatomists to finely delineate anatomical regions ^{6,7}. In contrast, atlases based on neuroimaging data

provide a 3D perspective of the brain with relatively lower resolution and simpler segmentations compared to histology based atlases^{11,18}. These atlases exist either in printed or online form (e.g., Waxholm space).”

- Page 1, line 21 ff:). the ARA ontology was adopted from Swanson et al. 2004 and Hof et al. 2000 (see AllenReferenceAtlas_v1_2008 Technical White Paper for details), if it is about the differences in brain region names the authors should discuss these references in more detail and give 2-3 examples (best with representative images)

To address this issue,

- 1) Figure 6 addresses this issue by inclusion of examples of different anatomical labels from both Allen and FP labels in the same area, with FP labels and Allen labels shown in white and black fonts, respectively.
- 2) We added the following text in the results while citing recommended references (page 7).
“The Allen labels and associated ontology were created based on previous works with mice and rats^{38,56,57}.”
“For example, cingulate cortex, area 24b (A24b) in the FP labels matches the anterior cingulate area, dorsal part (ACA_d) in the Allen labels (Figure 6C). Moreover, the primary motor cortex is abbreviated as M1 in the FP and MOp in the Allen labels (Figure 6C) and the bed nucleus of stria terminalis as ST in the FP and BST in the Allen labels (Figure 6D).”
- 3) We also added the following to the discussion.
“We digitized anatomical labels from new vector drawings and organized our labels hierarchically according to the Allen ontology^{38,56,57} as a neuroinformatics tool.”
“Discrepancies between atlases can arise from variations in anatomical delineations by different neuroanatomists, different sample preparation, strains, and normal phenotypic variation.”

- Line 26 ff: what is different to the approach described by Hjernevik et al. 2007?

Hjernevik et al. 2007 established computational methods to generate a 3D atlas from 2D images. This previous work provided an important tool for investigating anatomical structures from many different 3D imaging modalities (e.g., PET or MRI). Here, we adapted FP based labels and added new segmentations including dorsal striatum segmentations in the Allen CCF space. We have added importation of FP labels based on 2D histological Nissl staining into the Allen CCF space, which is based on serial two-photon tomography. Due to this difference in image modalities, manual alignment of FP based labels was required, using the criteria described in the manuscript.

Another major difference, one that is critical to the creation of smooth anatomical boundaries at multiple length scales, is that our boundaries were drawn using a vector based tool that is based on reference points, rather than exported bitmap drawings as done in Hjernevik et al. 2007. Vector based maps are scale free and easily editable. We share our entire vector drawing (Supplementary File 3) to facilitate refinement of the anatomical labels.

- Line 28 ff: references missing, I actually doubt that; the ARA ccf v3 comes with >1300 structures (including parental and childs)

We toned down the sentence and added missing references in the introduction.

“because it represents one of the most popular adult mouse brain atlases with detailed segmentations, and because a huge body of prior research is based on FP labels^{12,18}.”

Material&Methods

- Page 9, line 29 ff: the procedure on how the authors created the 3D labels is not well written and the selection of reference slides (e.g. every 10th ARA) seems to be arbitrary – at least it is not further explained. What puzzles me most: what where the criteria to decide for the CCF or FP/AMBMC landmark/region boundary in a particular situation and who did that, a single person? An overview figure would be helpful to understand the workflow (c.f. page 5, Allen Mouse CCF white paper)

To clarify our method,

- 1) We created Figure S6 to explain our work flow.
- 2) The method section has been updated with more detailed information of our alignment process.
“We used “shade” from levels of background autofluorescence and “texture” from fine myelinated tracks in the Allen CCF to determine anatomical borders. The shade feature was useful for delineating subregions in the isocortex, the hippocampus, the hindbrain, and the cerebellum; the texture feature was useful in the ventral striatum and the medulla. One or two slides before and after in each 2D section were used to ensure the contiguity of 3D labels.”
- 3) The following text was added to the methods section (page 10), to specify who performed the alignment.
“The primary alignment of each label was performed by U.C., followed by a second and independent inspection by Y.K.”
- 4) We added following to methods (p 12) regarding our choice of using 100 µm spacing.
“We chose 100 µm z spacing to facilitate comparisons between the ARA and the Allen CCF because the ARA was created in coronal sections evenly spaced at 100 µm intervals³⁸.”

- Page 9, line 44: Registration details missing, rigid/affine etc., make the code available in order to allow other users to register their data sets and further improve the atlas. I think that it is the crucial part of the manuscript as the registration of the atlas plates will induce the largest error if it is not done well - meaning related to specific anatomical landmarks (e.g. through a landmark correspondence approach, see Ito et al. Stroke

2018 <https://www.ahajournals.org/doi/10.1161/STROKEAHA.118.021508>)

To facilitate the registration of other data sets and improvement of the atlas, we have added more detail regarding image registration, including accuracy measurements. We agree that sharing the registration code would benefit the broader community. Since Elastix is publically available open source software, we added our registration parameter files for affine and bspline transformation as supplementary file 4.

See the following additions to the methods section on page 12.

“Image registration was performed using 3D image stacks at 20 µm x 20 µm x 50 µm (x,y,z) pixel spacing. Previously, we assessed the warping accuracy using 13 unique 3D landmarks acquired from Waxholm space¹⁰ in 6 different mouse brains after warping each dataset onto the reference brain²³. Warping accuracy was 65.0 ± 39.9 µm after 3D registration²³, supporting the accuracy of our registration results. Elastix associated files with detailed affine and bspline parameters can be found in supplementary file 4.”

Minor comments:

Introduction

- FP and Allen are quite uncommon abbreviations, especially Allen could be mistaken for Paul G. Allen, the founder of the Allen Institute – use ARA instead

We have updated the text throughout the manuscript to reflect use of Allen Reference Atlas (ARA) and Allen CCF to refer to the two different atlases released from Allen Institute for Brain Sciences in 2008 and 2017, respectively.

- Line 16: the 3D atlas is also particularly useful for *in vivo* imaging (PET, MRI, etc) see references to my comment on line 5

We have now added mention of *in vivo* imaging in the text below:

“This new reference brain marks a significant departure from classical neuroanatomy based on 2D sections and provides an excellent platform for the registration of 3D mouse brain imaging datasets collected from ***in vivo* imaging (e.g., PET, MRI)** and emerging high resolution whole brain imaging modalities such as serial two-photon tomography and light sheet microscopy”

- Typos: line 9 stainings, 19 delineations, 17 sharing of...I stop counting the errors made with the plural s, there are just too many. Revise the whole manuscript for typos.

We have corrected typos and grammar issues throughout the manuscript.

Reviewer #3 (Remarks to the Author):

In this manuscript, Chon et al. reports a valuable effort of developing a “translatable” 3D mouse brain atlas. Firstly, the authors registered anatomical labeling of two commonly used mouse brain atlases, Franklin and Paxinos (FP) and Allen Reference Atlas (Allen), into the same 3D Allen Common Coordinate Framework (CCF), allowing users to directly visualize and compare anatomical delineations and nomenclature of the mouse brain in the same 3D space frame. Secondly, they used cell type specific transgenic mice and an MRI atlas to adjust and further refine anatomical delineations. Thirdly, they refined the dorsal striatum (CP) into multiple subdivisions based on cortico-striatal connectivity data of the Allen Connectivity Database. Lastly, they have digitized the anatomical labels based on the Allen Reference Atlas ontology, created a web-interface for visualization, and provided tools for comprehensive comparisons between the Allen and FP labels.

We value the reviewer 3's accurate summary of our manuscript.

This study is a valuable exploration approach to address one fundamental problem in the field of neuroanatomy: how to address discrepancies among different brain atlases - for example, the FP and Allen, by constructing a unified mouse brain atlas. In the last decade, along with the rapid development of microscopic and computational technologies, digital brain atlases have become an effective informatics tool to facilitate data analysis and visualization, in addition to their classic role in referential texts. The Allen Reference Atlas (Dong, 2008) was the first digital atlas to serve this role and facilitate the development of the Allen Brain Atlas gene mapping project, Allen Connectivity Atlas, and other large-scale brain projects. The current study for the first time converts the FP delineations into a 3D-brain frame thereby allowing it to be directly compared with Allen delineations. This effort will be very well-received by the neuroscience community. Overall, the manuscript is well-written, and all figures are of high quality. Therefore, I strongly support its publication in Nature Communications.

We appreciate reviewer's enthusiasm.

My major comment is about the comparison of the FP and Allen. The FP, Allen CCF, Allen Reference Atlas, and MRI atlas are all in different imaging modalities and at different resolutions. First, it will be very useful for the authors to generate a table to show a detailed comparison of which structures are well-matching (let's say, >80% overlapping) between each of those three atlases (FP, Allen CCF, and Allen Reference Atlas) and how many structures display large discrepancies (i.e., < 50% overlapping?).

“We extended our analysis to the remaining brain regions and measured the degree of overlap between the three atlases at different ontological levels (Figure S7, Table S3). We performed Dice Similarity Coefficient (DSC) analysis to compute the degree of overlap between any given two atlases (1 = perfect overlap, 0 = no overlap). This approach provides a measure of the discrepancies between the atlases. Higher order structures showed overall good overlap (DSC over 0.8) while lower order structures showed pronounced discrepancies (DSC less than 0.5). For instance, DSC between Allen CCF and our FP based labels is 0.97 in the isocortex, 0.74 in the motor cortex, and 0.49 in the primary motor cortex. In contrast, overlap between the ARA and our labels is 0.95 in the isocortex, 0.80 in the motor cortex, and 0.69 in the primary motor cortex (Figure S7). A complete set of comparisons is provided in Table S3.”

We added Figure 7 and Table S3 to describe these results.

The following was added to the methods section (page 14).

“Overlap calculation between atlases

We used our FP based labels, the Allen CCF, and the ARA (each at 100 μm z spacing) to calculate overlap between atlases. The Dice Similarity Coefficient (DSC) was computed for each individual anatomical region including areas in different hierarchical ontology orders across atlases. DSCs were derived from individual absolute volumes (a region in Atlas A and a matched region in Atlas B) as well as the overlap of these volumes as Boolean data using overlap as true positive (TP; overlap), false positive (FP; A but B), and false negative (FN; B but A) between each pair of atlases as listed in Table S3. $DSC = 2TP / (2TP + FP + FN)$.”

Second, from my view, some of those discrepancies between the FP and Allen or between the Allen CCF and original Allen Reference Atlas are obviously due to different anatomical delineations from different neuroanatomists. However, some mismatching of labeling might due to different manners of tissue preparation and imaging process. The FP atlas was constructed using a brain from a mouse with 4% PFA perfusion (which may cause shrinkage of brain tissue); while the original Allen Reference Atlas was constructed with a fresh-frozen brain. Therefore, some mismatching labeling might be due to non-linear deformation of the brain tissues. The authors may want to discuss this issue and potential solution to reduce these discrepancies. For example, whether it will reduce the discrepancy of those two sets of labeling if they can register the corresponding Nissl-stained coronal sections (not just anatomical labeling) of FP and Allen Reference Atlas together.

We agree that discrepancies between FP and Allen atlases can arise from differences in sample preparation. Co-registering Nissl based 2D sections from the two atlases would definitely help to fit one atlas to another. However, one of our goals is to fit FP based labels in the Allen CCF with autofluorescent background images from serial two-photon tomography imaging, not the Nissl based Allen Reference Atlas. Differences in image content between FP (Nissl) and the Allen CCF are highly problematic during image registration.

We therefore added the following to the discussion:

“Discrepancies between atlases can arise from variations in anatomical delineations by different neuroanatomists, different sample preparation, strains, and normal phenotypic variation. For instance, the ARA was based on a fresh-frozen brain while the FP atlas was based on a brain

fixed with 4% paraformaldehyde^{6,7}. While co-registration of Nissl stained sections from two different atlases can help to reduce the differences, different z sampling rates (e.g., 100 μ m for the ARA and 120 μ m for the FP atlas) and challenges caused by registering images from different signal content (e.g., Nissl vs autofluorescent background in the Allen CCF) make it difficult to merge two independent atlases in one spatial context. We resolved the issues by combining manual alignments and co-registration of MRI and marker brains.”

The authors may also want to discuss in more details about other technical challenges and potential solutions of constructing this kind of “unified ” brain atlas(es).

To address other technical challenges and solutions in the construction of “unified” atlases, we have added the following to the discussion:

" Resolving discrepancies observed between atlases is of growing importance to the future of neuroscience due to constantly changing parameters of neuroimaging and spatially-resolved functional characterization. Rapid progress in large scale cell-type and connectivity mapping as well as *in vivo* neural recording will provide valuable information to refine anatomical borders, especially in areas with less clear cytoarchitectonic features, towards an increasingly rich, unified atlas of the mouse brain. The presented segmentations contain potential errors in boundaries of fine structures due to limitations of currently available marker brains and connectivity datasets. On-going and future efforts to generate large scale cell-type and connectivity database will no doubt facilitate the refining and even further segmentation of anatomical labels^{5,13,55}. Moreover, potential sex differences in marker expression may exist, and need to be considered in future work."

“In addition, atlas template brains based on different imaging modalities (e.g., MRI) co-registered to the Allen CCF space will facilitate incorporation of neuroimaging data from the different imaging modalities. Such development can bring us closer to the ideal of allowing data from different imaging modalities to be seamlessly registered into a common atlas space in which detailed anatomical segmentations are available.”

My other comments are mainly focused on their delineations of the striatum based on Allen Connectivity data: First of all, a recent report by Hintiryan et al. (Nature Neuroscience, 2016) produced a comprehensive anatomical delineation with fine-detailed nomenclature of the dorsal striatum based on the USC Mouse Connectome Project. It appears that by comparison, these two sets of delineations (the new delineation proposed in this manuscript and the one in Hintiryan et al.) have certain discrepancies. Therefore, to complete the analytical rigor of this report, the author may want to also adopt the delineations of Hintiryan et. al. into the same space frame.

We agree with the reviewer’s suggestion to adopt the delineations from Hintiryan et. al. 2016 Nat Neurosci. We therefore

- 1) imported the detailed segmentation in the dorsal striatum from Hintiryan et. al. in the Allen CCF background. We also used our own downloaded data from the Allen connectivity and another dorsal striatum segmentation dataset based on cortico-striatal projection (Hunnicut et al., 2016, eLife) to further guide the new dorsal striatum segmentation.
- 2) updated Figure 4 to reflect this new segmentation.
- 3) added the following text in the result section (page 5).

“A recent study from the Mouse Connectome Project generated highly detailed CP segmentations based on discrete cortical-striatal projections in the ARA³⁵. We primarily used this dataset as well as our data from the Allen connectivity and another cortico-striatum projectome dataset from Hunnicutt et al.,³⁶ to finely segment the CP. In the anterior-posterior axis, the CP was divided into the rostral extreme (re, Bregma A/P between +1.8 and +1.3), rostral (r, +1.2 and +0.7), intermediate (i, +0.6 and -0.4), caudal (c, -0.5 and -1.8), and caudal extreme (ce, -1.9 and -2.4). The CP at each level was further divided by community and domain as sub-segmentations as originally proposed by Hintiryan et al.,³⁵. For example, the CPi, dm, dl represents the dorsolateral (dl) domain within the dorsomedial (dm) community in an intermediate level CP (red arrow in Figure 4D4).”

- 4) added the following text in the methods section (page 11):

“To import segmentation dataset from Mouse Connectome Project (Center for Integrative Connectomics, University of Southern California), we used the cortico-striatal map (<http://www.mouseconnectome.org/CorticalMap/page/map/5>) to add different projection data to the ARA. This website provides a way to overlay up to 10 different projection data sets in 12 different anterior/posterior positions. A set of cortico-striatal projection to distinct CP subregions was selected based on clustering data from Hintiryan et al.,³⁵ and was overlaid in the Allen CCF background for segmentations using Adobe Illustrator. Five different anterior/posterior CP levels were determined by overlap, separation, and lack of projection from specific cortico-striatal projections^{35,36}. CPi,dm,dl and CPi,dm,d were combined with CPi,dm,dt due to a high degree of in their projection patterns³⁵.”

I also suggest that the authors consider generating different nomenclature for the new subdivisions of the striatum since the current nomenclature of calling new subdivisions of the striatum with different cortical areas is very unclear. Figure 3 in Hintiryan et. al. (2016) provides an excellent example that can be referenced for the authors' consideration.

To adopt the nomenclature from Hintiryan et. al. (2016) as the reviewer suggested, the following text was added to the results section:

“The CP at each level was further divided by community and domain as sub-segmentations as originally proposed by Hintiryan et al.,³⁵. For example, the CPi, dm, dl represents the dorsolateral (dl) domain within the dorsomedial (dm) community in an intermediate level CP (red arrow in Figure 4D4).”

The authors should be aware that the vast majority injections of the Allen Connectivity data are relatively large and involved in multiple cortical areas –the projection pattern may not be “clean.” Creating parcellations of the CP and labelling them with particular cortical areas such as “ACA” “MOs” may cause confusion. Furthermore, the authors should be aware that each of these subdivisions receive inputs from multiple cortical areas. For example, axonal terminals from ACA and RSP are overlapping in the CP. Again, labeling that part of the CP as the “ACA” may be misleading. Finally, the ORB-vl subdivision does not look fully accurate to me (seems slightly too ventral).

We agreed with the reviewer's concern. We resolved this issue by updating our entire dorsal striatum segmentation mainly based on the Hintiryan et. al. dataset that came with more refined brain injections.

Reviewer #4 (Remarks to the Author):

Enhanced and Unified Anatomical Labeling for a Common Mouse Brain Atlas

The authors present their efforts to integrate and unify the Franklin and Paxinos (FP) mouse brain atlas and the Allen Institute's common coordinate framework (CCF). They provide the result of this integration open and freely available.

The authors have provided an excellent service to the community by working through a number of challenging issues in integrating two widely adopted mouse brain atlases. Using a combination of MRI, cell-type specific labeling and long-range connectivity data they provide the labels of the FP atlas in the CCF space. The result is a valuable contribution to spatial data integration for the neuroscience community.

We thank the reviewer for positive comments of our work.

However, a few items would improve the manuscript:

The discussion focuses more on the motivation (which really is in the introduction). I would recommend that the Discussion highlight the limitations and opportunities for future improvements. The text would do well to have a caveats discussion to recognize the challenges of integrating multiple modalities of data, challenges in ensuring boundaries correspond and estimates of the potential error and which brain region are most vulnerable to mapping errors. To address this valuable community-oriented perspective, we have added the following to the discussion:

“The next step is to resolve discrepancies observed between atlases. Rapid progress in large scale cell-type and connectivity mapping as well as *in vivo* neural recording will facilitate the refinement of anatomical borders, especially in areas with less clear cytoarchitectonic features, towards a unified atlas of the mouse brain. The presented segmentations contain potential errors in boundaries of fine structures due to limitations of currently available marker brains and connectivity datasets. On-going and future efforts to generate large scale cell-type and connectivity database will no doubt facilitate the refining and even further segmentation of anatomical labels^{5,13,55}. Moreover, potential sex differences in marker expression may exist, and need to be considered in future work. On-going and future efforts to generate large scale cell-type and connectivity data will no doubt generate valuable information to refine and further segment anatomical labels and to create sex-specific brain atlases^{5,13,55}. To facilitate such work, we are making all the data (including vector drawing for delineations) freely available to visualize and download via our website. In addition, atlas template brains based on different imaging modalities (e.g., MRI) co-registered to the Allen CCF space will facilitate incorporation of neuroimaging data from the different imaging modalities. Such development can bring us closer to the ideal of allowing data from different imaging modalities to be seamlessly registered into a common atlas space in which detailed anatomical segmentations are available.”

The other issue is the terminology. Occasionally, the Allen Institute for Brain Science Common Coordinate Framework is referred to simply as “the Allen”. I would suggest using “the Allen CCF” as a minimum.

We updated our manuscript with “the Allen CCF” to refer the latest Allen template brain released in 2017 and “Allen Reference Atlas (ARA)” to refer the originally anatomical label drawn in Nissl-stained sections.

REVIEWERS' COMMENTS:

Reviewer #1 (Remarks to the Author):

The authors have performed a detailed revision based on the comments, thereby addressing all questions in a correct and relevant fashion, and the manuscript has improved considerably. I support publication of this work.

Reviewer #2 (Remarks to the Author):

The authors responded to the questions/comments sufficiently. The manuscript was improved substantially and the additional information and supplementary materials will allow researchers to use/reproduce the work. Note: the website seems to be unstable, sometimes it is not accessible. Authors should check their web server.

Markus Aswendt

Reviewer #3 (Remarks to the Author):

The authors have addressed all of my concerns. No more questions to ask. This work will be a great contribution to the neuroscience community.

Reviewer #4 (Remarks to the Author):

The authors have satisfactorily addressed my concerns with the manuscript.

REVIEWERS' COMMENTS:

Reviewer #1 (Remarks to the Author):

The authors have performed a detailed revision based on the comments, thereby addressing all questions in a correct and relevant fashion, and the manuscript has improved considerably. I support publication of this work.

Thank you

Reviewer #2 (Remarks to the Author):

The authors responded to the questions/comments sufficiently. The manuscript was improved substantially and the additional information and supplementary materials will allow researchers to use/reproduce the work. Note: the website seems to be unstable, sometimes it is not accessible. Authors should check their web server.

Markus Aswendt

Thank you.

We confirmed that our website is stable and works robustly.

Reviewer #3 (Remarks to the Author):

The authors have addressed all of my concerns. No more questions to ask. This work will be a great contribution to the neuroscience community.

Thank you

Reviewer #4 (Remarks to the Author):

The authors have satisfactorily addressed my concerns with the manuscript.

Thank you